# Silicon single-photon avalanche diodes with nano-structured light trapping

Kai Zang [1], Xiao Jiang[2,3], Yijie Huo[1], Xun Ding[2,3], Matthew Morea [1], Xiaochi Chen[1], Ching-Ying Lu[1], Jian Ma[2,3], Ming Zhou[4], Zhenyang Xia[4], Zongfu Yu[4], Theodore I. Kamins [1], Qiang Zhang[2,3] & James S. Harris[1]

Silicon single-photon avalanche detectors are becoming increasingly significant in research and in practical applications due to their high signal-to-noise ratio, complementary metal oxide semiconductor compatibility, room temperature operation, and cost-effectiveness. However, there is a trade-off in current silicon single-photon avalanche detectors, especially in the near infrared regime. Thick-junction devices have decent photon detection efficiency but poor timing jitter, while thin-junction devices have good timing jitter but poor efficiency. Here, we demonstrate a light-trapping, thin-junction Si single-photon avalanche diode that breaks this trade-off, by diffracting the incident photons into the horizontal waveguide mode, thus significantly increasing the absorption length. The photon detection efficiency has a 2.5-fold improvement in the near infrared regime, while the timing jitter remains 25 ps. The result provides a practical and complementary metal oxide semiconductor compatible method to improve the performance of single-photon avalanche detectors, image sensor arrays, and silicon photomultipliers over a broad spectral range.

[1] Department of Electrical Engineering, Stanford University, Stanford, CA 94305, USA. [2] Hefei National Laboratory for Physical Sciences at the Microscale and Department of Modern Physics, University of Science and Technology of China, Hefei, Anhui 230026, China. [3] CAS Center for Excellence and Synergetic Innovation Center in Quantum Information and Quantum Physics, Shanghai Branch, University of Science and Technology of China, Hefei, Anhui 230026, China. [4] Department of Electrical and Computer Engineering, University of Wisconsin-Madison, Madison, WI 53706, USA. Kai Zang and Xiao Jiang contributed equally to this work. Correspondence and requests for materials should be addressed to K.Z. (email: kaizang@stanford.edu)

A high-efficiency, low-jitter silicon single-photon avalanche detector (Si SPAD) has been desired for many applications, such as Lidar imaging[1–3], non-line-of-sight imaging[4], quantum communication and computation[5], bio-spectroscopy, in vivo molecular imaging[6], and medical imaging[7]. In an imaging system, SPADs with higher photon detection efficiency (PDE) are better for collecting photons, which decreases measurement time and ensures better signal-to-noise ratio. However, SPADs with lower jitter have an improved temporal resolution, which corresponds to a finer depth resolution in time-of-flight applications. A thin-junction, blue-shifted SPAD has a decent timing jitter of 35 ps full width at half maximum (FWHM)[8–11]. Yet due to the low absorption in the thin layer, such an SPAD has a low PDE compared to a thick-junction device[12]. For example, the peak PDE for a thin-junction SPAD is around 50% at 550 nm, while the typical PDE for a thick-junction SPAD is around 70% at 650 nm[8, 13]. Moreover, in all these applications, near-infrared wavelengths are often used to take advantage of fiber compatibility, eye-safety, and low absorption in solutions. However, typical PDEs of thin-junction SPADs at 850 and 980 nm are only 10% and 2% respectively[10].

Previous efforts to increase PDE above 850 nm wavelength come with a sacrifice in jitter distribution. Thick silicon is used to extend photon absorption regions in three ways (see Supplementary Fig. 1 for schematic illustration): extension of the avalanche region[12], extension of the depletion region to drift carriers toward the avalanche region[14], or extension of the neutral region[11]. The first two methods broaden the jitter, while the last one significantly extends the tail (i.e., an increased full width at 1% maximum in the jitter distribution) due to the slow diffusion process. An alternative solution to improving PDE is using a resonant cavity to create an optical resonance in the vertical direction, as in double-SOI-substrate, resonant-cavity-enhanced (RCE) SPADs[11]. However, the sharp resonances and low injection-angle tolerance[15] narrow their applications.

On the other hand, nano-structured materials have demonstrated excellent photon management and have been used in photodetectors (PDs)[16], avalanche photodetectors[17] and solar cells[18]. Although black silicon has been applied as a broadband anti-reflection layer to enhance absorption, especially for solar cells, devices still need thick silicon layers; thus, this method cannot provide a lower jitter[18, 19]. Metal nano-particles could couple photons into a light-trapping mode[20], but the metal itself will potentially induce a large loss.

Here, we demonstrate a light-trapping thin-film Si SPAD[21] that breaks the current trade-off between jitter and PDE. Compared to previous works where PDE follows a $1 - e^{-\alpha L}$ or $1 - e^{-2\alpha L}$ relationship with the device layer thickness $L$ and absorption coefficient $\alpha$, our design diffracts the vertically incident photons into a horizontal waveguide mode, and thus traps the photons in the thin-film to enhance PDE while keeping a low timing jitter[22].

## Results

**Device fabrication.** The light-trapping SPAD is a typical mesa-type, shallow-junction SPAD fabricated using a complementary metal oxide semiconductor (CMOS) compatible process. Devices with different diameters were built. Figure 1 shows three-dimensional (3D) structures and scanning electron microscope (SEM) images for 50 μm diameter SPADs as an example. Si epitaxial layers with a total thickness of 2.5 μm are grown on an SOI substrate. To compare nano-structured devices with control SPADs, a portion of the devices on the same wafer have nano-structures etched into the mesa surface. The nano-structure is etched as an inverse pyramid, with 850 nm period in a square lattice pattern. Thermal oxide of 100 nm thickness on the sidewall and the top serves as a passivation layer and a weak guard ring (due to dopant segregation at the oxide interface[23]). After fabrication, both the light-trapping and control SPADs have dark current as low as 40 fA at −1 V bias for 20 μm diameter devices, and ideality factor around 1.05 at 0.6 V. The breakdown voltage is 8.5 V at room temperature, corresponding to a 280 nm avalanche region. The breakdown voltage is lower than the design for two reasons. One is dopant diffusion during thermal oxidation at 1000 °C, which reduces the depletion region thickness (see Supplementary Fig. 2); the other is due to edge effects and lack of a guard ring; the breakdown probability is measured to be 20% higher on the edge.

**Light trapping SPAD performance simulation.** Finite-difference time-domain (FDTD) optical simulation is performed to

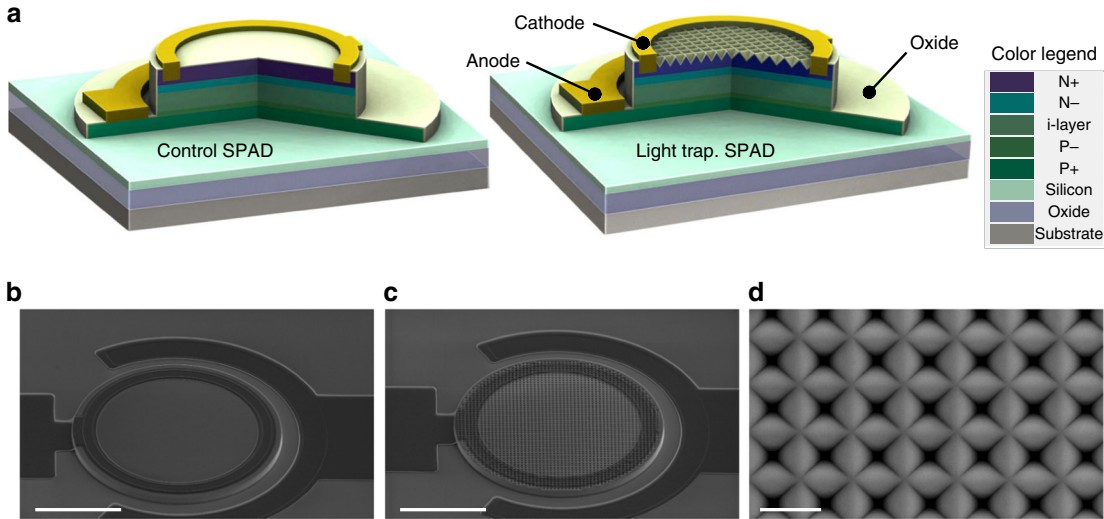

**Fig. 1** Structure of light-trapping SPAD. **a** Three-dimensional (3D) cross-sectional schematics of layer configurations of control (*left*) and light-trapping SPADs (*right*). The color legend shows the names of all the layers in both devices. The design thicknesses of layers from N+ to the bottom oxide layer of SOI substrate are sequentially listed 600, 300, 700, 200, 500, 220, and 400 nm for both devices. **b**, **c** Scanning electron microscope (SEM) images (45° view of *top* of structure) of control (**b**) and light-trapping SPADs (**c**) both with 50 μm diameter. *Scale bar* in both, 20 μm. **d** SEM image (*top-down view*) of inverse pyramid nano-structure on light-trapping SPAD. *Scale bar*, 1 μm

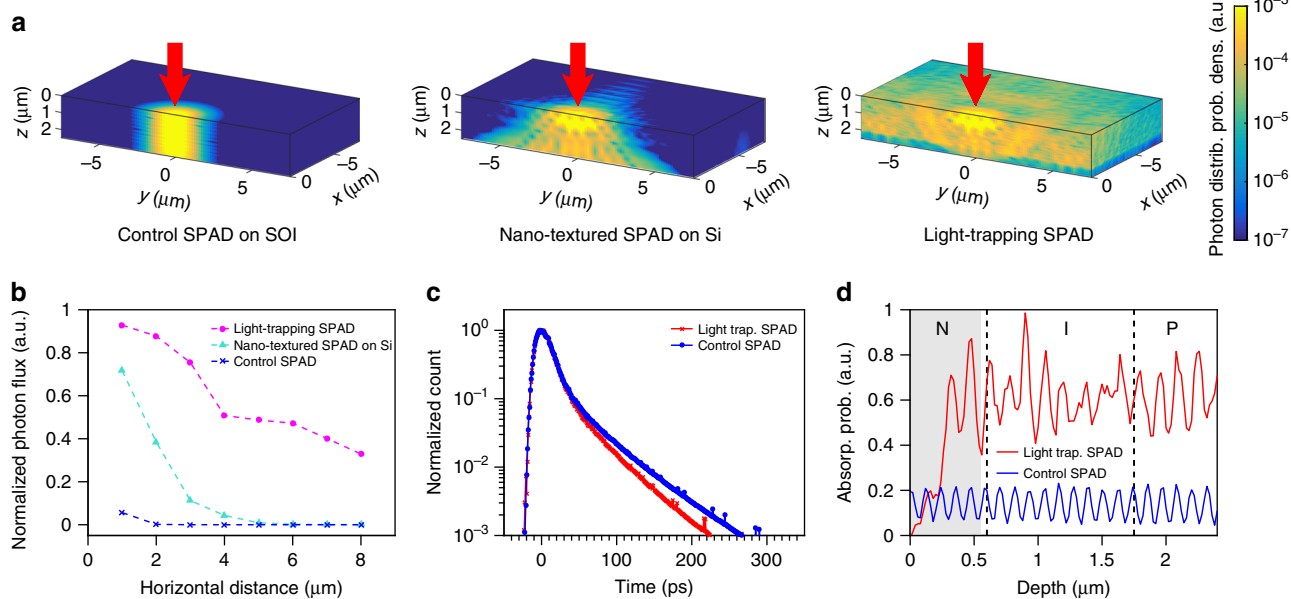

**Fig. 2** Simulation of photon distribution probability and jitter performance. **a** FDTD simulation of 850 nm wavelength photon distribution in a control SPAD (*left*), nano-textured SPAD on Si substrate (*middle*), and light-trapping SPAD (*right*). Photons are incident in positive Z direction. The *color bar* on the right shows the *color map* of photon distribution probability density. **b** Normalized flux of 850 nm photons propagating varying horizontal distance in a control SPAD (*blue cross*), nano-textured SPAD on Si substrate (*turquoise triangle*) and light-trapping SPAD (*magenta circle*). **c** Monte Carlo simulation of jitter distribution at given photon absorption distribution for a control SPAD (*blue*) and light-trapping SPAD (*red*). **d** Simulated photon absorption at varying depths for a control SPAD (*blue*) and light-trapping SPAD (*red*), with top surface at 0 µm. *Shaded areas* correspond to regions being nano-structured. *Dashed lines* denote respective doping layer in an NIP junction from *left* (surface) to *right* (substrate)

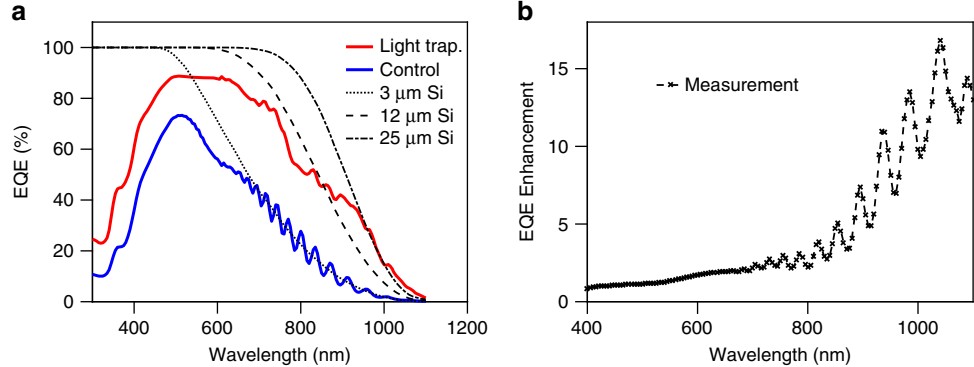

**Fig. 3** EQE measurement. **a** EQE measurements of 500 µm diameter control (*blue solid*) and light-trapping SPADs (*red solid*); *dashed black lines* correspond to theoretical absorption of 3, 12, and 25 µm thick Si from *left* to *right*. **b** EQE enhancement: ratio of light-trapping SPAD EQE compared to control SPAD EQE at different wavelengths

demonstrate the working principle of the light-trapping SPAD. A Gaussian beam of 850 nm wavelength is simulated to impinge vertically upon the surface of three different structures, namely a plane SPAD on an SOI substrate (control SPAD), a nano-textured SPAD on a Si substrate and a nano-textured SPAD on an SOI substrate (light-trapping SPAD). The horizontally propagating photon flux is simulated and recorded. As the results in Fig. 2a, b show, for the control SPAD, photons travel vertically with minimal horizontal component of the Poynting vector from the Gaussian beam. For the nano-textured SPAD on Si, the nano-structures provide a horizontal Poynting vector, but most of the photons travel through the device layer into the substrate due to lack of bottom reflection. For the light trapping SPAD, photons are diffracted by the nano-structures and maintain horizontal momentum by waveguiding between the bottom reflecting substrate and the air–Si interface. This increases absorption and

greatly broadens the photon distribution in the device layer, as shown in Fig. 2a.

In addition to enhanced absorption, this waveguiding effect breaks the trade-off between PDE and jitter, because the region of enhanced absorption overlaps the depletion region. A simulation of jitter distribution has been performed to make a qualitative comparison, assuming uniform electric field in the 1.2 µm thick depletion region. The electric field is chosen to be $4.5 \times 10^5$ V cm$^{-1}$, which corresponds to a FWHM jitter of 25 ps and an excess voltage that is 35% of the breakdown voltage. The simulation conditions are set to match our experimental result (25 ps, 30% excess-breakdown voltage ratio). As seen in Fig. 2c, the simulated jitter distribution of the control and light-trapping SPADs are almost identical. The similarity of FWHM jitter comes about because most of the photons are absorbed in depletion regions of similar thickness. However, the full-width tenth maximum

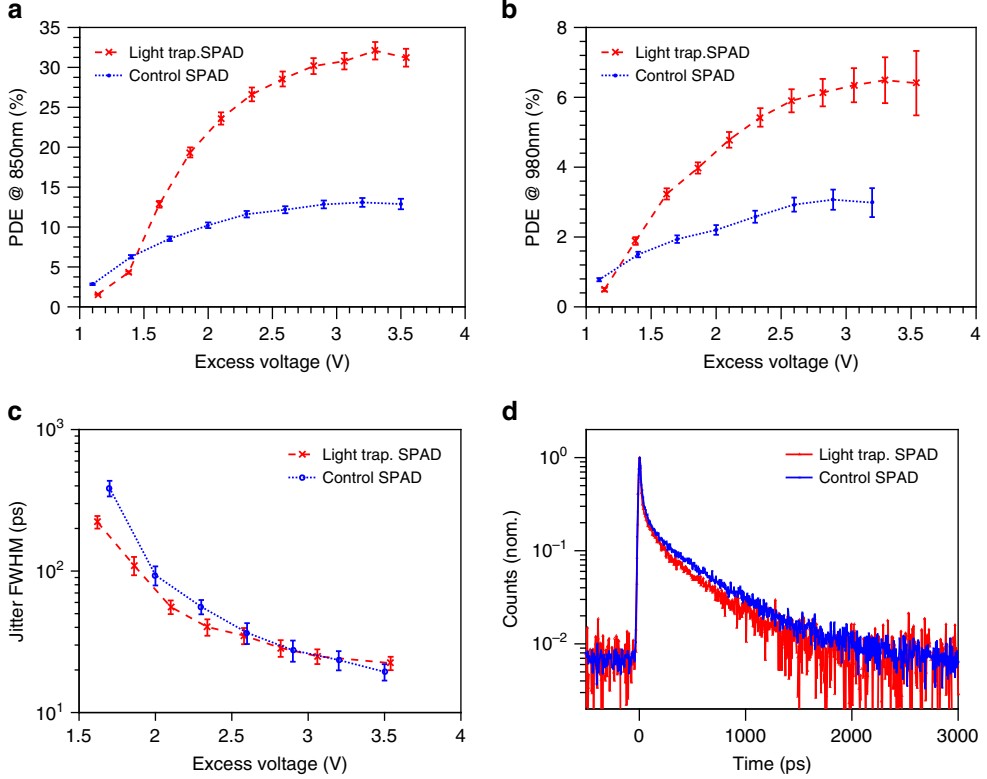

**Fig. 4** Single photon measurements. In all figures, *blue* stands for control SPADs and *red* for light-trapping SPADs. The *error bars* in **a**–**c** are estimated measurement uncertainty (see Supplementary Notes 3 and 4 for uncertainty analysis). **a** PDE vs. excess voltage at 850 nm wavelength, *error bar*, s. d. **b** PDE vs. excess voltage at 980 nm wavelength, *Error bar*, s.d. **c** Jitter (FWHM) vs. excess voltage at 940 nm and 1 photon per pulse, *error bar*, s.d. **d** Jitter distribution of two SPADs at 940 nm wavelength with FWHM of 25 ps, measured at 1 photon per pulse and corrected for pile-up effect

(FWTM) and tail of the light-trapping SPAD are slightly better than those of the control SPAD. This is because in light trapping SPADs, nano-structures are formed by etching silicon away, so fewer photons are absorbed in the top layer, as shown in Fig. 2d.

In our design, the top layer is heavily phosphorus doped to make holes the minority carriers. In Si, holes have a lower ionization probability compared to electrons. So chances for photo-generated carriers in the top layer to trigger an avalanche are smaller, and, therefore, they contribute less to the jitter distribution. The tail difference is expected to be larger if we reverse the doping polarity.

**Light trapping SPAD characterization.** External quantum efficiency (EQE) is used to evaluate absorption in an SPAD, which is defined as the ratio of photo-generated carriers collected by an SPAD to the number of external incident photons of a given wavelength. Measurements of EQE at 0 V confirm our simulation results, as shown in Fig. 3a. Unlike the resonance peaks found in the responsivity of RCE detectors, the light-trapping SPAD has broadband enhancement. This can be explained by the photons being efficiently coupled to the guided mode and being thus less dependent on constructive or destructive optical interference. At wavelengths below 700 nm, the effective absorption length is less than 5.2 μm, so that the incident photon is only reflected from the bottom interface once. Therefore, the enhancement of EQE mainly comes from the anti-reflection effect of the nano-structures (see Supplementary Note 1 and Supplementary Fig. 3), with an additional contribution from nano-structure diffraction. For wavelengths whose absorption is near the surface, the enhancement stems from, on average, less diffusion distance from the surface to the depletion region. Above 700 nm, the

waveguiding effect begins to dominate because photons have a greater absorption length. The dashed line in Fig. 3a shows that the EQE of the control SPAD follows the theoretical absorption of a 3 μm thick Si layer. However, the light-trapping SPAD has threefold improvement in EQE at 850 nm wavelength, equivalent to an effective absorption length of 12 μm, and tenfold improvement at 980 nm with an effective absorption length of 25 μm. This is further confirmed in Fig. 3b as the EQE enhancement shows a super-linear trend, which implies that light-trapping is more significant at wavelengths where Si absorption is low. The measured EQE of light-trapping SPADs is also insensitive to light incident angle (see Supplementary Note 2 and Supplementary Fig. 4).

With our demonstration of high absorption in a thin layer, we next want to verify the low jitter and high PDE properties of such a structure in the Geiger mode[24–26]. Geiger-mode measurements are carried out in a set-up with a gated quenching configuration (10 ns width, 1 kHz frequency) and one photon per pulse, as detailed in the Methods section. First, as shown in Fig. 4a, b, PDE is enhanced from 13 to 32% at 850 nm wavelength, and from 3 to 6.5% at 980 nm, which is consistent with the EQE results. This corresponds to a breakdown probability around 60%.

Regarding jitter, the light-trapping SPAD follows a very similar FWHM vs. excess voltage trend to that of the control SPAD, as shown in Fig. 4c for 940 nm wavelength. The smallest FWHM achieved is about 25 ps. This suggests that the jitter FWHM is still predominantly determined by the depletion thickness in a light-trapping SPAD. Figure 4d confirms that at a specific over-bias, the jitter distributions have an identical peak, with FWHM of 25 ps, which means that the majority of absorption is still in the depletion region. By contrast, light trapping SPADs have a

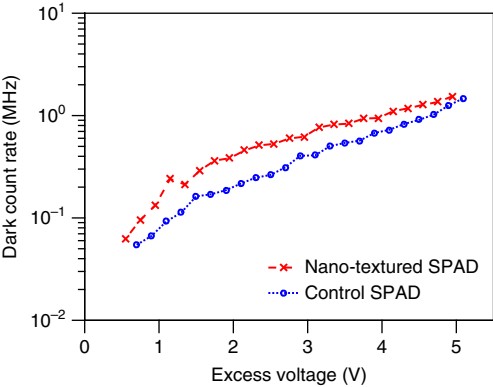

**Fig. 5** Dark count rate. Dark count rate comparison of thicker-junction SPADs (3.5 µm) on Si substrates with and without nanostructured surface

shorter tail compared to control SPADs or standard SPAD products, as predicted in the aforementioned simulations. At 940 nm wavelength, we achieved a jitter FWTM of 265 ps and tail (FW1%M) of 1380 ps for light-trapping SPADs compared to a FWTM of 335 ps and tail of 1755 ps for control SPADs.

## Discussion

The dark count rate (DCR) for both light trapping and control devices is around 40 MHz. This may come from epitaxy on SOI substrates, lack of a guard ring structure, and tunneling in the thin-junction[27, 28]. To determine if nano-structures affect the DCR at a lower level, we reduced the DCR by two orders of magnitude through fabricating thicker-junction SPADs, both nano-structured and control devices, on a Si substrate, without the light trapping effect. In Fig. 5, nano-textured SPADs have 2–3 times higher DCR than their control SPADs, which implies that the DCR is degraded but remains within the same order of magnitude. Atomic layer deposition of $Al_2O_3$ and standard guard ring are suggested to further improve the DCR of light trapping SPADs[18].

The light trapping SPAD makes absorption and avalanche regions overlap and be confined in a small thickness. Given better passivation for surface nanostructures, the dead time and after-pulsing will remain small compared to thick-junction SPADs. SPAD designs without slow tails have been demonstrated with compromised PDE[29]; light trapping therefore improves PDE for better-performing, tail-free SPADs. The surface texturing process is CMOS and lithography compatible, making it easier to be integrated in current SPAD fabrication processes and easier to transfer to SPAD image sensors and silicon photo-multipliers (SiPMs). In addition, it can also help enhance photon number counting in SiPMs, where photons will be diffracted, guided, and absorbed in different pixels, even if they impinge on the same pixel, especially for a closely patterned SiPM[30]. In SPAD image sensors, on the other hand, this guiding effect may lead to optical cross-talk. But it could be solved by using deep trench isolation with metal sandwiched by dielectrics, so that photons reflect at the boundary with minimum absorption, but at the cost of a more complicated fabrication process.

In conclusion, we demonstrated a CMOS-compatible method to surface texture SOI-based SPADs, which helps to break the trade-off between jitter and PDE while not seriously deteriorating other SPAD performance metrics. Because of light trapping, PDE is enhanced by approximately 2.5 times at the wavelengths of 850 and 980 nm, while jitter is still as low as 25 ps FWHM. This method could also be transferred to CMOS-compatible SPAD image sensors[31] or Si Photo-multiplier arrays[30].

## Methods

**Device fabrication**. A 2.5 µm silicon device layer is first epitaxially grown in a reduced pressure chemical vapor deposition reactor on an SOI substrate with 220 nm Si on a 400 nm buried-oxide layer. Then, 200 nm low-temperature oxide (LTO) is deposited in a low-pressure chemical vapor deposition reactor at 400 °C as a hard mask for etching nano-structures. Square or hexagonal patterned holes with 850 nm period are opened in the LTO mask layer by lithography and plasma etching using $NF_3$ gas. The entire wafer is then dipped in tetramethylammonium hydroxide at 70 °C to wet etch silicon and form 600 nm deep etched inverse pyramids as nano-structures. After surface texturing, device mesas of different diameters are plasma etched in an HBr, $Cl_2$, and $O_2$ gas environment, followed by cleaning and thermal oxidation at 1000 °C for passivation. Fabrication is completed with HF wet etching to open vias and metal deposition of 10 nm Ti and 200 nm Al.

**Single-photon measurement set-up**. The set-up is configured to measure PDE, which is denoted as the probability of photon detection per pulse with only one incident photon, and jitter, which is the distribution of photon detection time at avalanche threshold. PDE and jitter are measured simultaneously. Photons at 895 nm wavelength and in a 3 ps pulse from a femtosecond laser (Coherent Mira900) are attenuated and used as a single-photon light source. The photons are then collimated to a large Gaussian spot, with diameter of 2.7 mm, and projected on the small SPAD under test with center alignment. Attenuation is adjusted based on light spot size and device active area. The SPAD is put into a translational and rotational stage, and its electrical response is recorded by a wideband oscilloscope (Agilent DSO91204A) with timing resolution better than 2.5 ps. Gating voltage source (Agilent 81150) and oscilloscope are all synchronized to the femtosecond laser pulse in a gated quenching configuration. During the testing, the gating rate is 1 kHz, derived from the laser pulse repeat rate of 76 MHz. The photon number per pulse is monitored at the single-photon level. The avalanche pulse is recorded by the oscilloscope, with a discriminating threshold of 50 mV, and the PDE and jitter performance are estimated by analyzing the recorded pulses. The probability of multi photons per pulse according to Poisson distribution has been considered and corrections have been used during data processing to get the same PDE and jitter performances as for a strict single-photon test light pulse, even at high photon flux.

**Data availability**. The data that support the findings of this study are available from the corresponding author upon reasonable request.

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

## Acknowledgements

This work was supported by the National Fundamental Research Program (under Grant No. 2013CB336800), the Chinese Academy of Science, and the National Natural Science Foundation of China. Device fabrication and characterization was performed in part in the nano@Stanford labs, which are supported by the National Science Foundation as part of the National Nanotechnology Coordinated Infrastructure under award ECCS-1542152. Z.F.Y. acknowledges the financial support by DARPA DETECT program. M.Z. is partially supported by NSF Grant No. ECCS-1641006. We especially thank Prof. Jian-Wei Pan for his guidance and insightful discussions.

## Author contributions

Q.Z. and J.S.H. conceived and supervised the entire project. K.Z., X.J., Y.J.H., Q.Z., and J.S.H. designed the experiments. K.Z. and Y.J.H. performed epitaxy and device fabrication. T.I.K. assisted the design of the epitaxial and fabrication processes. M.M., X.C.C., and C.Y.L. helped fabrication. X.J., X.D., and Q.Z. characterized the devices. J.M., M.Z., and Z.F.Y. simulated device performance. Z.F.Y. and T.I.K contributed to data analysis. X.J. and Z.Y.X. prepared the figures. K.Z., X.J., Y.J.H., and Q.Z. drafted the manuscript, and all co-authors contributed to and proofread the manuscript.

## Additional information

**Competing interests:** The authors declare no competing financial interests.

