## [Peer Review File · Nature Communications]

Reviewers' Comments:

Reviewer #1 (Remarks to the Author)

This paper reports the performance improvement of silicon single photon avalanche detector (SPAD); here the highlight is to employ a surface nano-texturing in order to utilize light trapping and reduce surface reflection. By doing so, they could get similar timing jitter but much enhanced photon detection efficiency (PDE).

This approach was firstly proposed by the same group in 2015 (J. Ma et al, Optica); and now they present the fabrication and characterization of the devices.

Even though the idea (nanostructuring for light trapping) has been applied in other research fields such as solar cells

the reported results are likely to be of high interest in scientific community.

However, the authors need to address or comment on the following point.

In practical viewpoints nanostructuring of the surface might degrade the device performance, dark current in particular.

In addition the waveguiding effect of texturing would be detrimental effect in the crosstalk between neighbouring pixels

in imaging array for example. Though the authors suggest deep trench isolation with metal sandwiched by dielectrics,

this results in more complicated fabrication process.

Reviewer #2 (Remarks to the Author)

The authors present a method for increasing photon absorption in Single-Photon Avalanche Diodes (SPAD) while keeping the active volume small in order to preserve low timing jitter. In detail, a nano-structured layer is etched on the photon entry surface and a light trapping is created between such nano-structured surface and the underlying SOI layer, with the active volume in between.

This idea is novel and could be of interest to SPAD designers.

Indeed, this paper shows that the proposed approach should work, but further (more accurate) data are needed to prove the enhancement in the absorption efficiency.

Here are specific comments:

- The authors fabricated a mesa-type device, with 100 nm oxide layer on the walls: I think that this passivation layer does not provide any "guard ring" effect. Could you better explain the guard ring mentioned on page 3, line 58-59?

- Page 4: The breakdown voltage is very low, showing that the electric field in the depleted region is quite high, thus leading to tunneling effects. It is even lower than the one simulated by the authors during the design: is this due to edge effects? Did the authors acquire a measurement of the electroluminescence emitted by the detector in Geiger-mode for checking if breakdown probability is uniform all over the active area or is higher on the edge?

- Page 4: How was the timing jitter simulated? In which conditions was it simulated? (e.g. what was the excess bias voltage?)

- At which bias was the external quantum efficiency measured?

- On page 5: the authors identified the wavelength of 700 nm as a limit below which the EQE enhancement is mainly due to the antireflection effect, while above it the enhancement is mainly due to the waveguiding effect. Can you better justify this analysis? Where does the "700 nm limit" come from? How does this limit depend on the nano-structured pattern?

- On page 6 and Fig. 4: the authors mention that the decrease of PDE at high excess bias voltages is due to higher DCR and leakage currents. I agree that this can be the case for the acquired raw

data, with maybe an additional contribution from the afterpulsing effect, but it has to be properly considered and corrections have to be applied. The authors should better explain how these measurements were acquired and processed.

- In jitter distribution measurements, a tail of 800 ps (is this the time constant?) is reported, which is quite long (not "small" as reported in the manuscript). How do you explain it? What is its origin?
- The measured DCR is very high (40 MHz) and cannot be compared to that of other SPADs fabricated in completely different technology (a planar 130 nm CMOS technology). Why do you compare this DCR to that of ref. 23?
- The DCR is so high that it may mask any subtle difference between the light trapping SPAD and the control one, thus not proving that the nano-structured pattern is not affecting DCR.
- On page 7, the authors state that this approach "can reduce dead time, after-pulsing and the tail as well." Can you explain how?
- Fig. 4d: If the jitter distribution is measured at 6.5 photons per pulse impinging the SPAD, multiplying by the PDE we have on average about 1 avalanche per pulse, meaning that the SPAD count rate is saturated and the reconstructed waveform is distorted. Please, explain how jitter distribution measurements were acquired and in which conditions.
- Supplementary information, page 1: the angular dependence of EQE was measured with an integrating sphere, but its output is not a collimated beam and photons are emitted with various angles. Why did you choose an integrating sphere for this measurement? Do you have an estimation of the characteristics of the optical beam output by the sphere?
- Supplementary information, page 2, Fig. S2 a: reflection measurements were acquired from a sample with a 700 nm period of the nano-structured pattern, while all the other measurements were from a sample with 850 nm period. Why did you change the pattern period? Do you have these measurements from the sample with 850 nm period?

Minor comments:

- References 1 and 2 are not papers, but links to commercially-available detectors. The authors should refer to some of the many papers published on journals in this field.
- Page 2, line 25: Please, review this sentence: "From system performance, higher PDE often refers to ..."
- Page 2, line 30: "single-photon counting module (SPCM)" is a commercial name of a product. These detectors are usually called "thick-junction SPAD", as opposed to "thin-junction SPAD".
- Page 2, line 33: at least one reference is needed here for supporting the cited numbers.
- Page 2, line 35-36: the authors should better describe the three ways for collecting photons by properly referring to the differences among the three regions (e.g. in terms of electric field).
- Page 4, line 67: what do you mean with "monitors"?
- Page 6, line 102: when Geiger-mode is cited, the authors refer to "ref. 22". I suggest revising this reference since references that are more appropriate are available in literature.
- Page 6, line 104: At which excess bias were the measurements of Fig. 4 acquired?
- Page 7, line 123: Please, revise this sentence "Thus it not only improves timing and PDE, ..."
- Page 8, line 158: How long is the gate?
- Supplementary information: page 1, fig. S1 b: the measured data are pretty noisy, are they repeatable?
- The authors should revise the manuscript for correcting typos and English errors.

Reviewer #3 (Remarks to the Author)

This paper has some interesting results regarding enhancement of detector efficiency using nanostructures. However, the quality of the writing is poor, and some descriptions are just plain wrong. I also have strong reservations about the validity of the Geiger mode measurements.

In the abstract "However, Si SPADs suffer from a challenging trade-off between timing jitter and photon detection efficiency (PDE)" This is incorrect, there is no straightforward trade-off between

PDE and jitter – for a given device geometry, the PDE and jitter both increase with bias, at the expense of dark count rate. On the other hand, if a thick, efficient homojunction is designed, it may adversely affect jitter slightly. Is this what the authors mean? That is difficult to say, as the point they are making is lost with the lack of explanation.

A high efficiency, low jitter silicon SPAD has been desired for many applications, such as Lidar imaging [7, 8], quantum communication and computation [9], bio-spectroscopy, in vivo molecular imaging [10] and medical imaging [11]. First of all, reference 8 is not really an imaging paper in the normal sense of the description and is better substituted by a paper by the same group the Garipey et al. Nature Commun. 6, 6021 (2015). Depth imaging using silicon single-photon detectors is better represented by a review such as IEEE J. Sel. Top. Quant. Electron. 13, 1006–1015 (2007)

“while smaller jitter is less variation in temporal or spatial domain of imaging system.” Should state lower jitter corresponds to improved temporal resolution, resulting in improved depth resolution” or similar. Use of the phrase “spatial domain” is vague and confusing.

“A thin film, blue-shifted SPAD has a decent timing jitter of 35 ps at full width half maximum (FWHM).” Where is the reference for this work?

“Thick silicon is used to collect photo-carriers in three ways: directly in the depletion region [12], through drift of carriers into the depletion region [13], or with diffusion of carriers into the depletion region [14].” Again, this is badly worded and very confusing. Photo-generated carriers drift in the depleted region towards the multiplication region, or they diffuse into the depleted region and then drift towards the multiplication region. Carriers can’t drift into the depletion region as they are not under an electric field. This process is explained in a number of relevant reviews: Vol. 35, Issue 12, pp. 1956-1976 (1996), or Meas. Sci. Technol. 21 012002 (2010)”

“In our design, the top layer is heavily phosphorus doped with holes as minority carriers and with a lower breakdown probability compared to electrons.” This is confusing, are the authors trying to say that there is a different avalanche breakdown for holes and electrons? This is either a badly written sentence or a very deep misunderstanding of the physics of the impact ionization process.

“External Quantum Efficiency (EQE)” What exactly does this mean? Is this single-photon detection efficiency, or conversion of incident photons to electron-hole pairs?

“At wavelengths below 700 nm, the enhancement mainly comes from the anti-reflection effect of the nano-structure, with an additional contribution from nano-structure diffraction and effective collection of carriers.” I am not sure I understand why the carrier collection is more efficient when the nanostructure is present?

Figure 4 appears to show single-photon detection efficiency. These results have some potential from the point of view of improvement between devices, but I am very concerned that an SPDE of up to 25% is observed when a pulse containing 6.5 photons per pulse is incident. This surely will lead to highly distorted results as time-correlated single photon counting must operate only in the regime of low probability of detection, typically below 5% probability of a detection event when compared to the incident pulse rate. Otherwise, this will lead to highly unreliable SPDE and jitter results. (See “Pile-Up Effect” in W. Becker “Advanced Time-Correlated Single Photon Counting Techniques” Springer 2005). Essentially, the SPDE is the probability of a single-photon being measured, it is not the probability of a detection event if multiple photons are incident – that probability will always be higher than the SPDE. Generally, such an SPDE measurement will be valid if the probability of detection is very low, so a combination of high detection efficiency and large number of photons per pulse cannot be valid conditions for an SPDE measurement.

The DCR is 40MHz, which is exceptionally high for a silicon device, and will make measurement of SPDE very difficult. This is not mentioned. In fact, groups have shown even Ge-on-Si SPADs

without guard rings with lower DCR than that described, (IEEE JOURNAL OF QUANTUM ELECTRONICS, VOL. 47, NO. 5, MAY 2011 and IEEE TRANSACTIONS ON ELECTRON DEVICES, VOL. 60, NO. 11, NOVEMBER 2013). Of course, to achieve this, these groups cooled the detector and this allowed accurate SPDE measurements at a low photon number per pulse. These groups electrically gated the device during testing to reduce the overall measured count rate – was this done here? One highly plausible reason that the dark count rate of these all-Si devices are so high is that the structure is grown on SOI and not on an all-silicon substrate – an issue not discussed in this manuscript. Was an equivalent control sample grown on an all-silicon substrate measured?

In conclusion this is an interesting manuscript, with evidence of efficiency enhancement by use of the nanostructure, and will merit interest from the single photon community if placed in a well-written manuscript. However, the paper is badly written, does not place the work in context, does not cite other work well and has poor description of device operation. The information provided along with the results of Figure 4 makes the measurement look incorrect. The conditions of high DCR and high photon number per pulse can make the efficiency result look higher than they actually are, and is a common mistake made by researchers new to the subject. I would suggest that the group cool the detectors to reduce the dark count rate and use a much lower number of photons per pulse. Doing this will give the community confidence that the results are correct. In addition, there are a number of other omissions in the manuscript (described above) that need attention. Overall, this paper is lacking adequate description of the measurement techniques and device physics. The Geiger mode results are, at best, questionable despite the possible enhancement seen in performance. I cannot recommend publication in its present form and suggest the authors revisit some of the Geiger-mode measurements, or direct a heavily edited manuscript in a different manner.

Reply to Reviewer #1

Comment 1:

This paper reports the performance improvement of silicon single photon avalanche detector (SPAD); here the highlight is to employ a surface nano-texturing in order to utilize light trapping and reduce surface reflection. By doing so, they could get similar timing jitter but much enhanced photon detection efficiency (PDE). This approach was firstly proposed by the same group in 2015 (J. Ma et al, Optica); and now they present the fabrication and characterization of the devices. Even though the idea (nanostructuring for light trapping) has been applied in other research fields such as solar cells, the reported results are likely to be of high interest in scientific community.

However, the authors need to address or comment on the following point.

1) In practical viewpoints, nanostructuring of the surface might degrade the device performance, dark current in particular.

Reply:

We would like to thank the reviewer to think our work "of high interest in scientific community". We follow the reviewer's suggestion to implement measurements of dark current and dark count rate (DCR) of the nano-texturing SPAD. By comparison, SPAD dark currents remain almost the same after nano-texturing. SPAD DCRs are, however 2-3 times higher after nano-texturing yet remain within the same order of magnitude. Detailed measurement results are shown below.

In Fig C-1, IV measurements show that dark currents for 20 μm diameter SPADs are about the same in the low-reverse-bias regime, for both the nano-texturing and control (without nano-texturing) SPAD. Under higher voltages, the nano-textured SPAD does induce a very small increase in dark current and a small increase in breakdown voltage. The slight shift is observed in SPADs fabricated on both Si substrates and SOI substrates.

The reasons that dark currents remain the same after nano-structuring, are in part due to 1) thermal oxidation passivation, 2) use of TMAH solutions to wet etch

nanostructures, and 3) heavy phosphorus doping in the top layer. First, thermal oxidation process induces the dopant diffusion, and pushes the depletion region away from surface nanostructures. This can be verified in the doping profile simulation as in Fig C-5. Thermal oxidation also passivates SPADs better than low-temperature oxide (LTO) and has a 30-fold decrease in dark current. Second, TMAH solution etches Si and stops on the crystal lattice (111) plane, which helps reduce recombination centers. Lastly, the heavy phosphorus doping in the nano-structures decreases the minority carrier diffusion length. To further prevent degradation, we expect that an atomic layer deposition (ALD) of Al_2O_3 thin film will help. ALD of Al_2O_3 is a relatively easy and low cost approach to reduce surface recombination of nano-textured surface. This has been demonstrated in Ref [1] with black silicon solar cells, whose efficiency has dramatically increased with help of Al_2O_3 ALD.

Figure C-1 Dark current measurements for light trapping (nano-structured surface) and control SPADs.

For the DCR test, we have fabricated another batch of thicker-junction SPADs ($\sim 3.5 \mu\text{m}$) on a Si substrate with and without nanostructures and compared their DCR. In Fig C-2, the light trapping SPADs have about 2-3 times more DCR than the control SPADs. The results show that, nanostructures wet etched by TMAH solution degrade SPAD DCR but remain within the same order of magnitude, on a 100 kHz scale.

Again, we suppose that using Al_2O_3 ALD will help decrease DCR further as explained above.

Figure C-2 Dark count rate comparison of thicker junction SPADs with and without nanostructured surface.

To make this point clear, we revise the manuscript as below:

Line 70: "After fabrication, both the light-trapping and control SPADs have dark current as low as 40fA at -1V bias for $20\mu\text{m}$ diameter devices, and ideality factor around 1.05 at 0.6V."

Line 138: "To determine if nano-structures affect the DCR at a lower level, we reduced the DCR by two orders of magnitude through fabricating thicker-junction SPADs, both nano-structured and control ones, on a Si substrate, without light trapping effect. In Fig 6, nano-textured SPADs have 2-3 times higher DCR than their control SPADs, which implies that the DCR is degraded but remains within the same order of magnitude. Atomic layer deposition (ALD) of Al_2O_3 and standard guard ring are suggested to further improve DCR of light trapping SPADs."

Comment 2:

2) In addition the waveguiding effect of texturing would be detrimental effect in the crosstalk between neighboring pixels in imaging array for example. Though the authors suggest deep trench isolation with metal sandwiched by dielectrics, this results in more complicated fabrication process.

Reply:

We thank the referee for this valuable comment. We agree that metal sandwiched by dielectrics will complicate the process. At the same time, dielectric DTI has been used in CMOS image sensor prototypes recently ([2], [3]). This paves way to dielectric-metal-dielectric sandwiched DTI structure, a techniques that is slightly complicated but feasible.

In order to make this point clear, we revise the manuscript as below:

Line 153: "In SPAD image sensors, on the other hand, this guiding effect may lead to optical cross-talk. But it could be solved by using deep trench isolation with metal sandwiched by dielectrics, so that photons reflect at the boundary with minimum absorption, but at the cost of a more complicated fabrication process."

Below we further explain why DTI process is preferred.

Figure C-3 Photon distribution at a wavelength of 905 nm after being diffracted by 850 nm period nanostructures. Black arrows represent the first three orders of diffraction based on 1D grating theory.

In Fig C-3, 3D simulations reveal that the diffraction of photons roughly follows the grating theory. The critical angle for total internal reflection between Si and SiO₂ is 66.2°, which suggests that by simply using oxide DTI the majority of photons will be reflected back instead of penetrating to the adjacent pixels. This means that the crosstalk will be limited, even with a dielectric-filled DTI. A dielectric-metal-dielectric sandwich DTI will thus further help reduce optical crosstalk.

Reply to Reviewer #2

Comment 1:

The authors present a method for increasing photon absorption in Single-Photon Avalanche Diodes (SPAD) while keeping the active volume small in order to preserve low timing jitter. In detail, a nano-structured layer is etched on the photon entry surface and a light trapping is created between such nano-structured surface and the underlying SOI layer, with the active volume in between. This idea is novel and could be of interest to SPAD designers. Indeed, this paper shows that the proposed approach should work, but further (more accurate) data are needed to prove the enhancement in the absorption efficiency.

Here are specific comments:

- The authors fabricated a mesa-type device, with 100 nm oxide layer on the walls: I think that this passivation layer does not provide any "guard ring" effect. Could you better explain the guard ring mentioned on page 3, line 58-59?

Reply:

We thank the reviewer for thinking our work "novel and could be of interest".

During the thermal oxidation process, dopant diffusion and segregation at the oxide interface leads to an accumulation of phosphorus and depletion of boron at the sidewall [4], which results in reduced electric field at the sidewall junction. This leads to a guard ring effect, which is rather weak compared to standard techniques, so SPAD DCR remains high. Fig C-4 shows the device simulation result of electric field distribution within the depletion region at a reverse bias of 10 V and an explanation of the weak guard ring effect.

In order to make this point clear, we revise the manuscript as below:

Line 69: "Thermal oxidation of 100nm thickness on the sidewall and top serves as a passivation layer and a weak guard ring (due to dopant segregation at the oxide interface)."

Figure C-4 Thermal oxide serves as a “weak” guard ring. **(a)** Electric field distribution when biased at -10 V. The simulated SPAD has a PIN junction as designed and goes through a thermal oxidation step. Subplot figure is the enlarged view on the mesa sidewall. **(b)** A schematic to explain why the electric field distribution within the depletion layer is different between the sidewall and the mesa center. The dashed grey lines outline the depletion region.

Comment 2:

- Page 4: The breakdown voltage is very low, showing that the electric field in the depleted region is quite high, thus leading to tunneling effects. It is even lower than the one simulated by the authors during the design: is this due to edge effects? Did the authors acquire a measurement of the electroluminescence emitted by the detector in Geiger-mode for checking if breakdown probability is uniform all over the active area or is higher on the edge?

Reply:

We thank the reviewer for this comment. The low breakdown voltage is due to two reasons: dopant diffusion during thermal oxidation and edge effects.

The first issue, dopant diffusion, comes from the thermal oxidation process at 1000 °C. Fig C-5 plots the dopant distribution before and after the oxidation process. After the thermal oxidation, there is a higher doping in intrinsic region. So the depletion width decreases, which reduces the SPAD breakdown voltage.

Figure C-5 Simulated epitaxial doping profile before and after thermal oxidation

Figure C-6 Photon detection efficiency (PDE) uniformity over a 50 μm diameter light trapping SPAD. Color bar with higher number denotes higher PDE with 850 nm wavelength light focused at that spot. Each grid in the figure corresponds to $\sim 1.5 \times 1.5 \mu\text{m}^2$ area.

For the second issue, we measured PDEs with a focused light spot and scanned across a light trapping SPAD. Results in Fig C-6 prove the existence of edge effects. PDE on the edge is $\sim 20\%$ higher than that in the center, which implies more avalanche events occur in regions close to SPAD edge.

Comment 3:

- Page 4: How was the timing jitter simulated? In which conditions was it simulated? (e.g. what was the excess bias voltage?)
- At which bias was the external quantum efficiency measured?

Reply:

The timing jitter simulation follows the method of Ref [7] in the depletion region, using Monte Carlo based random path length (RPL) models. The tail of the jitter distribution follows the method of Ref [8], using diffusion equations to model carrier dynamics in the neutral region. We assume that the top 600 nm layer (N+) and bottom 600 nm layer (P-) are neutral layers and that the middle 1.2 μm layer is the depletion region with a uniform electric field. The above assumption does not reflect the actual doping in fabricated SPADs; rather, they serve as a qualitative measure to understand performance comparison between the light-trapping and control SPADs.

We simulate SPAD jitter distributions at 4.5×10^5 V/cm, where the jitter FWHM matches 25 ps as measured. Given that Si avalanche breakdown electric field is 3.35×10^5 V/cm, this corresponds to an excess voltage that is 35% of the breakdown voltage (40.2 V). This breakdown voltage is much higher than the experimental results, because the depletion width is assumed much larger and is free from dopant diffusion. However, excess voltage versus breakdown voltage ratio of 35% matches well with experimental results ($\sim 30\%$) for both light trapping and control SPADs.

Meanwhile, external quantum efficiency (EQE) was measured at 0 V.

In order to make this point clear, we revise the manuscript as:

Line 103: "Measurements of External Quantum Efficiency (EQE) at 0V confirm our simulation results, as shown in Fig. 4(a)."

Comment 4:

- On page 5: the authors identified the wavelength of 700 nm as a limit below which the EQE enhancement is mainly due to the antireflection effect, while above it the enhancement is mainly due to the waveguiding effect. Can you better justify this analysis? Where does the "700 nm limit" come from? How does this limit depend on the nano-structured pattern?

Reply:

We thank the reviewer for pointing out this unclear point. In Si, 700 nm wavelength corresponds to an absorption length of $5.2 \mu\text{m}$, which is about double the thickness of the device layer. Highly likely, photons of below 700 nm wavelength will be absorbed before finishing "one round trip" in Si layers. This can be verified in Fig. 3(a) EQE measurement of the revised manuscript, where interference pattern emerges after 700 nm wavelength in control SPADs.

Photons that travel longer are more likely for wavelengths larger than 700 nm due to lower absorption coefficients in Si. As in Fig 3(a), the characteristic propagation length for light-trapping SPADs at 850 nm wavelength is around $12 \mu\text{m}$.

This has nothing to do with nanostructure patterns, since it only depends on Si absorption lengths of different wavelengths.

We revise the manuscript as below:

Line 107: "At wavelengths below 700 nm, the effective absorption length is less than $5.2 \mu\text{m}$, so that the incident photon is only reflected from the bottom interface once. Therefore, the enhancement of EQE mainly comes from the anti-reflection effect of the nano-structures, with an additional contribution from nano-structure diffraction. For wavelengths whose absorption concentrates on the surface, the enhancement stems from on average less diffusion distance from the surface to the depletion region. Above 700nm, the waveguiding effect begins to dominate because photons have a greater absorption length."

Comment 5:

- On page 6 and Fig. 4: the authors mention that the decrease of PDE at high excess bias voltages is due to higher DCR and leakage currents. I agree that this can be the case for the acquired raw data, with maybe an additional contribution from the afterpulsing effect, but it has to be properly considered and corrections have to be applied. The authors should better explain how these measurements were acquired and processed.

Reply:

We highly appreciate the reviewer's comment. After careful debugging, we have to admit that the decrease of PDE at very high excess voltage is mainly due to our characterization negligence. After correction, the observed PDE reaches saturation when increasing excess voltages. The results have been updated in Fig 5 in the revised manuscript. As shown in the Appendix I, we also conduct simulations and experiments to confirm that the high DCR or multiple photons per pulse will not affect PDE results using our method.

Below we would like to explain how we obtained the incorrect result. In characterization, we use a gated quenching configuration with 10 ns gate width at 1 kHz repetition rate. A digital storage oscilloscope is used to capture SPAD responses in a 6.4 ns monitoring window out of the 10 ns gate width. Then, a data processing program is used to digitally discriminate whether an avalanche occurs in the captured response, by cross-threshold method. There are three possible scenarios to be captured by the oscilloscope, as shown in Fig C-7. These are (a) no avalanches within the monitoring window or the avalanches occur after the monitoring window; (b) an avalanche occurs right in the monitoring window; (c) an avalanche occurs before the monitoring window. In our previous program, scenario (c) was not discriminated from scenario (a), and they were both treated as "no trigger" in this detection period. As a result the measured PDE decreases when scenario (c) increased dramatically at very high excess voltages.

Figure C-7 Three possible scenarios when gate voltage is ON and an oscilloscope monitors only a portion of the gate time. (a) No avalanches or avalanches happen after the monitoring window. (b) Avalanches happen during the monitoring window. (c) Avalanches happen before the monitoring window.

Afterpulsing should not be a concern since we use a gated quenching configuration. With repetition rate of 1 kHz, afterpulsing is measured to be minimal after 1 ms dead time.

A detailed explanation on how PDE has been measured and a proof of measurement validity can be found in the Appendix I: PDE characterization, at the end of this letter.

Comment 6:

- In jitter distribution measurements, a tail of 800 ps (is this the time constant?) is reported, which is quite long (not "small" as reported in the manuscript). How do you explain it? What is its origin?

Reply:

We thank the reviewer for pointing this out. Tail of a SPAD is not the time constant. Here, it refers to full width at 1% maximum (FW1%M) of SPAD jitter distribution. For a standard SPAD, like SPCM, the tail is around 2 ns. For blue shifted SPAD from PMD,

the tail is generally more than 1 ns. That's why we claim our tail is "small". The tail that is longer than simulation, is coming from the distorted doping profile due to dopant diffusion during the thermal oxidation process. This can be verified in Fig C-4. After the oxidation process, depletion width decreases and neutral region width expands, so it takes longer for the photo-generated carriers to reach the avalanche region.

We have redone jitter distribution measurement and updated the wording in the revised manuscript as below:

Line 131: "Further, light trapping SPADs have a shorter tail compared to control SPADs or standard SPAD products, as predicted in the aforementioned simulations. At 940 nm wavelength, we achieved a jitter FWTM of 265 ps and tail (FW1%M) of 1380 ps for light-trapping SPADs compared to a FWTM of 335 ps and tail of 1755 ps for control SPADs."

Comment 7:

- The measured DCR is very high (40 MHz) and cannot be compared to that of other SPADs fabricated in completely different technology (a planar 130 nm CMOS technology). Why do you compare this DCR to that of ref. 23?

Reply:

We thank the reviewer for this comment. Ref. [23] in the original manuscript adopts the shallow trench isolation (STI) techniques as a guard ring, which is different from the traditional guard ring design by tuning doping profile outlined in Ref [9]. DCR shown in the reference is on the same order as our sample. However, we agree that it is not a fair comparison and should be removed, so we adjusted the manuscript accordingly.

Line 136: "The dark count rate (DCR) for both light trapping and control devices is around 40MHz. This may come from epitaxy on SOI substrates, lack of a guard ring structure, and tunneling in the thin-junction."

Comment 8:

- The DCR is so high that it may mask any subtle difference between the light trapping SPAD and the control one, thus not proving that the nano-structured pattern is not affecting DCR.

Reply:

We appreciate the reviewer for this valuable comment. We agree that the high DCR may influence the difference between the nano-texturing and control SPAD. Therefore, we re-fabricate a batch with nanostructures on a Si substrate and deliberately increase the epitaxy thickness. In this way, DCR is reduced to 10 kHz – 1 MHz. Although strong light trapping is missing in these newly fabricated thicker-junction samples, we believe that they provide a reasonable measure of the influence of the nanostructures. In Fig. C-2 below, nano-textured SPADs have about 2-3 times more DCR than control SPADs. The results show that, nanostructures degrade SPAD DCR but remain within the same order of magnitude, at least on a 100 kHz scale. To further prevent degradation, we expect that an atomic layer deposition (ALD) of Al_2O_3 thin film will help. ALD of Al_2O_3 is a relatively easy and low cost approach to reduce surface recombination of nano-textured surface. This has been demonstrated in Ref [1] with black silicon solar cells, whose efficiency has dramatically increased with help of Al_2O_3 ALD.

Fig. C-2 is also added to the revised manuscript as Fig. 6. And the manuscript is revised as:

Line 138: "To determine if nano-structures affect the DCR at a lower level, we reduced the DCR by two orders of magnitude through fabricating thicker-junction SPADs, both nano-structured and control ones, on a Si substrate without light trapping effect. In Fig. 6, nano-textured SPADs have 2–3 times higher DCR than their control SPADs, which implies that the DCR is degraded but remains within the same order of magnitude. Atomic layer deposition (ALD) of Al_2O_3 and standard guard ring are suggested to further improve DCR of light trapping SPADs."

Figure C-2 Dark count rate comparison of thicker junction SPADs with and without nanostructure surface.

Comment 9:

- On page 7, the authors state that this approach "can reduce dead time, after-pulsing and the tail as well." Can you explain how?

Reply:

Using the light-trapping method, the avalanche region can be confined to a smaller volume as opposed to thick-junction SPAD design. Given better passivation techniques for the nanostructures (e.g. Al₂O₃ ALD in Ref[1]), afterpulsing and dead time can be reduced compared to thick-junction SPADs.

An SPAD with a negligible tail has already been demonstrated [10]; however, it suffers from low PDE in the near infrared regime. This is because only the carriers absorbed in the depletion region will avalanche, while carriers absorbed in the

neutral region is blocked. Light trapping would be an ideal way to enhance its PDE performance.

The manuscript is revised as:

Line 144: "The light trapping SPAD makes absorption and avalanche regions overlap and be confined in a small volume. Given better passivation for surface nanostructures, the dead time and after pulsing will remain small compared to thick-junction SPADs. SPAD designs without slow tails have been demonstrated with compromised PDE; light trapping therefore improves PDE and paves the way for better performing tail-free SPADs."

Comment 10:

- Fig. 4d: If the jitter distribution is measured at 6.5 photons per pulse impinging the SPAD, multiplying by the PDE we have on average about 1 avalanche per pulse, meaning that the SPAD count rate is saturated and the reconstructed waveform is distorted. Please, explain how jitter distribution measurements were acquired and in which conditions.

Reply:

We thank the referee for this valuable comment. The measurement of jitter distribution presented in Fig. 4(d) of the original manuscript was based on an 895 nm wavelength femtosecond laser (Ti:sapphire) with a pulse width of 3 ps. A 10 GHz bandwidth, 40 GS/s sampling rate oscilloscope was used to discriminate the avalanche waveform with a sub-picosecond resolution. When PDE is 25%, the dark count rate was also about 25% trigger probability in a 6.4 ns period. To increase the signal to noise ratio (SNR) to a level that FWTM and FW(1%)M could be easily recognized from the jitter distribution, the femtosecond laser was attenuated to a level of 6.5 photons per pulse, which has a larger photon flux compared to normal SPAD tests. According to Poisson distribution, at a PDE of 25%, 6.5 photons per pulse corresponds to a trigger probability of about 80% per pulse and can produce a good SNR for FWTM and FW(1%)M measurements.

However, thanks for your reminder, we realize that the jitter distributions in Fig 4(d) of the original manuscript do suffer from the pile-up distortion. So we numerically

simulate how pile-up effects distort jitter distributions given our testing specifications, with the details in the Appendix at the end of this letter. The conclusions are that the FWHM suffers very little from pile-up distortion, but the FWTM and FW(1%)M do need pile-up correction if high photon flux is used for measurements. After pile-up correction, high photon flux can be used to measure jitter distribution.

To get the correct jitter distribution, we switch the laser wavelength to 940 nm, perform the pile-up correction algorithm and redo the measurements at 1 photon per pulse. At 940 nm wavelength, PDEs for both SPADs are lower (measured to be 15% for light-trapping SPADs and 5.5% for control SPADs). To demonstrate that pile-up correction method works, average 1, 6 and 20 photons per pulse were used to test the same light trapping SPAD. Fig C-8(a) shows the original measurement results, where pile-up distortions are easily observed, especially in the high photon flux scenarios. Then, by using a simple pile-up correction algorithm [11], the original jitter distribution is successfully restored, as shown in Fig C-8(b). The jitter distributions under 1, 6 and 20 photons per pulse overlap with each other quite well, which confirms that the pile-up correction works well and high photon flux could be used. A FWHM of 25 ps, FWTM of 265 ps, and tail of 1380 ps can be read from the corrected jitter distributions. We have updated jitter distributions in Fig. 5d of the revised manuscript, to the pile-up corrected ones, measured at 1 photon per pulse and at 940 nm wavelength.

Figure C-8 Pile-up effects in jitter distribution measurements and their corrections. **(a)** Raw data of measured jitter distributions of a light-trapping SPAD, affected by pile-up effect. **(b)** Jitter distributions after applying pile-up correction to the raw data in (a).

Explanations on jitter distribution measurements validity are located in the Appendix II: jitter distribution characterization, at the end of this letter. We have also revised the manuscript accordingly.

Line 126: "Regarding jitter, the light-trapping SPAD follows a very similar FWHM vs. excess voltage trend to that of the control SPAD, as shown in Fig. 5(c). The smallest FWHM achieved is about 25 ps. This suggests that the jitter FWHM is still predominately determined by the depletion thickness in a light-trapping SPAD. Fig. 5(d) confirms that at a specific over-bias, the jitter distributions have an identical peak, with FWHM of 25 ps, which means that the majority of absorption is still in the depletion region. Further, light trapping SPADs have a shorter tail compared to control SPADs or standard SPAD products, as predicted in the aforementioned simulations. At 940 nm wavelength, we achieved a jitter FWTM of 265 ps and tail (FW1%M) of 1380 ps for light-trapping SPADs compared to a FWTM of 335 ps and tail of 1755 ps for control SPADs."

Comment 11:

- Supplementary information, page 1: the angular dependence of EQE was measured with an integrating sphere, but its output is not a collimated beam and photons are emitted with various angles. Why did you choose an integrating sphere for this measurement? Do you have an estimation of the characteristics of the optical beam output by the sphere?

Reply:

Thanks for pointing it out. This was a typo. We used a sphere with black coating (which is different from an integrating sphere) inside simply to rule out ambient light and not to characterize optical beam output.

We have revised the first sentence in the first paragraph of Supplementary information as "The light trapping SPAD angular dependence on external quantum efficiency (EQE) has been measured in a rotational stage with the ambient light blocked."

Comment 12:

- Supplementary information, page 2, Fig. S2 a: reflection measurements were acquired from a sample with a 700 nm period of the nano-structured pattern, while all the other measurements were from a sample with 850 nm period. Why did you change the pattern period? Do you have these measurements from the sample with 850 nm period?

Reply:

Thanks for your comments. Samples with an 850 nm period are characterized in the manuscript and their reflections are shown below in Fig C-9. It demonstrates good anti-reflection, which limits reflection to below 5% in the near infrared regime. In addition, the result also shows a clear shift of resonance. However, at larger injection angles, the resonance wavelength is approaching the responsivity limit of Si photodiodes that we use. Therefore in the supplementary files, we used reflection results from the sample with 700 nm period nano-structures, which shows a clearer shift of resonance.

Figure C-9 Reflection measurement of 850 nm period inverse pyramid nano-structures on Si substrate at different injection angles.

Comment 13:

Minor comments:

- References 1 and 2 are not papers, but links to commercially-available detectors. The authors should refer to some of the many paper published on journals in this field.

Reply:

Thanks for the comment. We agree with you and update the reference with some reviews on SPADs in the revised manuscript. The new references are listed as below for the sake of convenience.

- Hadfield, R. H. Single-photon detectors for optical quantum information applications. *Nature Photonics* 3, 696–705 (2009).
- Buller, G. & Collins, R. Single-photon generation and detection. *Measurement Science and Technology* 21, 012002 (2009).
- Eisaman, M., Fan, J., Migdall, A. & Polyakov, S. Invited review article: Single-photon sources and detectors. *Review of Scientific Instruments* 82, 071101 (2011).
- Hall, D., Liu, Y.-H. & Lo, Y.-H. Single photon avalanche detectors: prospects of new quenching and gain mechanisms. *Nanophotonics* 4, 397–412 (2015).
- Ghioni, M., Gulinatti, A., Rech, I., Zappa, F. & Cova, S. Progress in silicon single-photon avalanche diodes. *IEEE Journal of Selected Topics in Quantum Electronics* 13, 852–862 (2007).

Comment 14:

- Page 2, line 25: Please, review this sentence: "From system performance, higher PDE often refers to ..."

Reply:

Thanks for pointing it out. We rephrased it in the revised manuscript as below:

Line 29: "In an imaging system, SPADs with higher PDE are more excellent in collecting photons, which decreases measurement time and ensures better signal-to-noise-ratio; while SPADs with lower jitter have an improved temporal resolution, which corresponds to a finer depth resolution in time-of-flight applications."

Comment 15:

- Page 2, line 30: "single-photon counting module (SPCM)" is a commercial name of a product. These detectors are usually called "thick-junction SPAD", as opposed to "thin-junction SPAD".

Reply:

Thanks for pointing it out. We have updated it accordingly in the revised manuscript.

Comment 16:

- Page 2, line 33: at least one reference is needed here for supporting the cited numbers.

Reply:

Thanks for pointing it out. We have added one reference below to the manuscript. The figure of merit was read from the curve of PDE vs. Wavelength in Fig. 4 of the reference.

- Ghioni, M., Gulinatti, A., Rech, I., Zappa, F. & Cova, S. Progress in silicon single-photon avalanche diodes. IEEE Journal of selected topics in quantum electronics 13, 852–862 (2007).

Comment 17:

- Page 2, line 35-36: the authors should better describe the three ways for collecting photons by properly referring to the differences among the three regions (e.g. in terms of electric field).

Reply:

We have included a new figure in the manuscript to better describe how to collect photons in different regions. The figure is copied here as Fig C-10.

Figure C-10 Schematics and electric field distributions for respective SPAD designs. Dashed lines denote depletion region boundaries. **(a)** Control device thin junction SPAD based on PN junction. **(b)** Absorption enhancement based on thicker avalanche region. **(c)** Absorption enhancement based on extended depletion region to drift carriers to the avalanche region. **(d)** Absorption enhancement based on extended diffusion region. **(e)** Resonant-cavity-enhanced structure.

And the second paragraph of the main text is revised as:

Line 40: "Previous efforts to increase PDE above 850 nm wavelength comes with a sacrifice in jitter distribution. Fig. 1(b-e) lists several methods to optimize the PDE of the thin-junction SPAD shown in Fig. 1(a). Thick silicon is used to extend photo absorption regions in three ways: extension of the avalanche region (Fig. 1(b)), extension of the depletion region to drift carriers towards the avalanche region (Fig. 1(c)), or extension of the neutral region (Fig. 1(d)). The first two methods will broaden the jitter, while the last one will significantly extend the tail (i.e., an increased full width at 1% maximum in the jitter distribution), due to the slow diffusion process. An alternative solution to improving PDE is using a resonant cavity to create an optical

resonance in the vertical direction (Fig. 1(e)), as in double SOI substrate resonant cavity enhanced (RCE) SPADs. However, the sharp resonances and low injection angle tolerance narrow their applications.”

Comment 18:

- Page 4, line 67: what do you mean with "monitors"?

Reply:

In FDTD simulations, “monitors” are used to measure the total photon flux transmission through a rectangular area specified by this “monitor.”

The manuscript is revised as:

Line 79: “The horizontally propagating photon flux is simulated and recorded.”

Comment 19:

- Page 6, line 102: when Geiger-mode is cited, the authors refer to "ref. 22". I suggest revising this reference since references that are more appropriate are available in literature.

Reply:

Thanks for the suggestion. We have cited more appropriate references on Geiger-mode operation and measurement in the manuscript. They are listed below for the convenience.

- Cova, S., Ghioni, M., Lacaíta, A., Samori, C. & Zappa, F. Avalanche photodiodes and quenching circuits for single-photon detection. *Applied Optics* 35, 1956–1976 (1996).
- López, M., Hofer, H. & Kück, S. Detection efficiency calibration of single-photon silicon avalanche photodiodes traceable using double attenuator technique. *Journal of Modern Optics* 62, 1732–1738 (2015).

- Acerbi, F. et al. Characterization of single-photon time resolution: from single spad to silicon photomultiplier. IEEE Transactions on Nuclear Science 61, 2678–2686 (2014).

Comment 20:

- Page 6, line 104: At which excess bias were the measurements of Fig. 4 acquired?

Reply:

In the original manuscript, excess voltage was 1.5 V because PDE decreases at higher excess voltage. This is due to a bug in data processing program as explained earlier. In the revised manuscript, the excess voltage is 2.8 V.

Comment 21:

- Page 7, line 123: Please, revise this sentence "Thus it not only improves timing and PDE, ..."

Reply:

This part is updated as follows: "Given better passivation for surface nanostructures, the dead time and afterpulsing will remain small compared to thick-junction SPADs. SPAD designs without slow tails have been demonstrated with compromised PDEs; light trapping therefore improves PDE and paves the way for better performing tail-free SPADs."

Comment 22:

- Page 8, line 158: How long is the gate?

Reply:

The gate width is around 10 ns, repeated at 1 kHz.

Comment 23:

- Supplementary information: page 1, fig. S1 b: the measured data are pretty noisy, are they repeatable?

Reply:

Thanks for your comment. Below we compare the results obtained recently with the data in the original manuscript (obtained 8 months ago). Fig C-11 shows consistent results. The large discrepancy at high injection angles may be from blocking or reflection of bonding wires in the original experiment and larger measurement uncertainties.

Figure C-11 EQE angle dependent measurement when ϕ is 0° for 850 nm and 980 nm wavelengths. Measurement uncertainties are plotted as error bars for the repeated measurement.

Comment 24:

- The authors should revise the manuscript for correcting typos and English errors.

Reply:

Thanks for your suggestion. We have reworked the manuscript.

Reply to Reviewer #3

Comment 1:

This paper has some interesting results regarding enhancement of detector efficiency using nanostructures. However, the quality of the writing is poor, and some descriptions are just plain wrong. I also have strong reservations about the validity of the Geiger mode measurements.

In the abstract "However, Si SPADs suffer from a challenging trade-off between timing jitter and photon detection efficiency (PDE)" This is incorrect, there is no straightforward trade-off between PDE and jitter – for a given device geometry, the PDE and jitter both increase with bias, at the expense of dark count rate. On the other hand, if a thick, efficient homojunction is designed, it may adversely affect jitter slightly. Is this what the authors mean? That is difficult to say, as the point they are making is lost with the lack of explanation.

Reply:

We thank the reviewer to think our manuscript "interesting". We agree that both PDE and jitter get better when increasing bias voltage. However, the trade-off we mentioned is raised when choosing the junction thickness of a SPAD, not the bias voltage. For better understanding, we have revised the abstract as:

Line 18: "However, there is a trade-off in current Si SPADs. Thick-junction SPADs have good photon detection efficiency (PDE) but poor timing jitter, while thin-junction SPADs have good timing jitter but poor PDE. Here, we design and demonstrate a light trapping thin-junction Si SPAD to break this trade-off..."

In this way, we hope the concept of trade-off could be better delivered. Below, we would like to give an in-depth explanation why this trade-off exists.

Firstly, for a given SPAD with increasing excess voltage, PDE will saturate to a point where it is limited by optical absorption. Similarly, with increasing excess voltage,

jitter distribution, especially full width at half maximum (FWHM), will decrease to an asymptotic value. The minimum for jitter FWHM, to first order, could be approximated as the transit time for carriers to drift across the depletion region at saturated velocity ($\sim 10^7$ cm/s for electrons in Si). In addition, jitter FWHM of thicker-junction SPADs tends to be even larger, because it suffers from other effects such as secondary photon emission or diffusion-assisted avalanche spreading from the space charge effect (described in Ref [12]).

Secondly, Si has poor absorption in the near-infrared regime. Therefore, higher PDE favors a thicker Si layer, while lower jitter FWHM favors a thinner Si layer. If SPADs operate at their optimal excess voltage, SPADs of different junction thickness will have a clear trade-off between PDE and jitter FWHM. Similar claims can be found in Ref [13].

Comment 2:

A high efficiency, low jitter silicon SPAD has been desired for many applications, such as Lidar imaging [7, 8], quantum communication and computation [9], bio-spectroscopy, in vivo molecular imaging [10] and medical imaging [11]. First of all, reference 8 is not really an imaging paper in the normal sense of the description and is better substituted by a paper by the same group the Gariepy et al. Nature Commun. 6, 6021 (2015). Depth imaging using silicon single-photon detectors is better represented by a review such as IEEE J. Sel. Top. Quant. Electron. 13, 1006–1015 (2007)

Reply:

We thank the reviewer for pointing out the inappropriate references. We have updated the references in the revised manuscript accordingly.

The original reference 8 is referred to as “non-line-of-sight imaging” in the revised manuscript. The two references you mentioned do provide great helps in revising this manuscript and planning our future work. They are added in the revised manuscript. They were overlooked because we only focused on the design of SPAD at that stage.

Comment 3:

“while smaller jitter is less variation in temporal or spatial domain of imaging system.” Should state lower jitter corresponds to improved temporal resolution, resulting in improved depth resolution” or similar. Use of the phrase “spatial domain” is vague and confusing.

Reply:

We thank the reviewer for this insightful comment. We agree that the phrase “spatial domain” is vague, especially in the field of imaging. The manuscript is updated as follows:

Line 29: “In an imaging system, SPADs with higher PDE are more excellent in collecting photons, which decreases measurement time and ensures better signal-noise-ratio; while SPADs with lower jitter have an improved temporal resolution, which corresponds to a finer depth resolution in time-of-flight applications.”

Comment 4:

“A thin film, blue-shifted SPAD has a decent timing jitter of 35 ps at full width half maximum (FWHM).” Where is the reference for this work?

Reply:

Thanks for the reminding. We notice that several references mention this performance. Some of them are listed below. And they are added to the revised manuscript.

- Eisaman, M., Fan, J., Migdall, A. & Polyakov, S. Invited review article: Single-photon sources and detectors. *Review of Scientific Instruments* 82, 071101 (2011).
- Hall, D., Liu, Y.-H. & Lo, Y.-H. Single photon avalanche detectors: prospects of new quenching and gain mechanisms. *Nanophotonics* 4, 397–412 (2015).
- Ghioni, M., Gulinatti, A., Rech, I., Zappa, F. & Cova, S. Progress in silicon single-photon avalanche diodes. *IEEE Journal of Selected Topics in Quantum Electronics* 13, 852–862 (2007).

- Ghioni, M. et al. Resonant-cavity-enhanced single photon avalanche diodes on double silicon-on-insulator substrates. *Journal of Modern Optics* 56, 309–316 (2009).

Comment 5:

“Thick silicon is used to collect photo-carriers in three ways: directly in the depletion region [12], through drift of carriers into the depletion region [13], or with diffusion of carriers into the depletion region [14].” Again, this is badly worded and very confusing. Photo-generated carriers drift in the depleted region towards the multiplication region, or they diffuse into the depleted region and then drift towards the multiplication region. Carriers can’t drift into the depletion region as they are not under an electric field. This process is explained in a number of relevant reviews: Vol. 35, Issue 12, pp. 1956-1976 (1996), or *Meas. Sci. Technol.* 21 012002 (2010)”

Reply:

We appreciate the reviewer for pointing out this issue, because we confused “depletion region” with “avalanche region” in the original manuscript. Within the depletion region, only electric fields larger than 3×10^5 V/cm will induce avalanches and multiplications. While it only requires $\sim 1 \times 10^5$ V/cm electric fields to get electrons in Si to reach saturation velocity. The difference is presented in Fig. C-10 (b) and (c).

Figure C-10 Schematics and electric field distributions for respective SPAD designs. Dashed lines denote depletion region boundaries. **(a)** Control device thin junction SPAD based on PN junction. **(b)** Absorption enhancement based on thicker avalanche region. **(c)** Absorption enhancement based on extended depletion region to drift carriers to the avalanche region. **(d)** Absorption enhancement based on extended diffusion region. **(e)** Resonant-cavity-enhanced structure.

The above Fig. C-10 is added to the revised manuscript as Fig. 1. And the 2nd paragraph of the main text is revised as below:

Line 40: "Previous efforts to increase PDE above 850nm comes with a sacrifice in jitter distribution. Fig. 1(b-e) lists several methods to optimize the PDE of the thin-junction SPAD shown in Fig. 1(a). Thick silicon is used to extend photo absorption regions in three ways: extension of the avalanche region (Fig. 1(b)), extension of the depletion region to drift carriers towards the avalanche region (Fig. 1(c)), or extension of the neutral region (Fig. 1(d)). The first two designs will broaden the jitter, and the last one will significantly extend the tail (i.e., an increased full width at 1% maximum in the jitter distribution), due to the slow diffusion process. An alternative solution to improving PDE is using a resonant cavity to create an optical resonance in the vertical direction (Fig. 1(e)), as in double SOI substrate resonant cavity enhanced (RCE)

SPADs. However, the sharp resonances and low injection angle tolerance narrow their applications.”

Comment 6:

“In our design, the top layer is heavily phosphorus doped with holes as minority carriers and with a lower breakdown probability compared to electrons.” This is confusing, are the authors trying to say that there is a different avalanche breakdown for holes and electrons? This is either a badly written sentence or a very deep misunderstanding of the physics of the impact ionization process.

Reply:

Thanks for your comments. Avalanche breakdowns created by either holes or electrons are the same but the probability associated with them is different. In Si, electrons are more likely to be ionized than holes at the same electric field. So the device breakdown probability P_{BD} (the probability a photon successfully triggering an SPAD) is not uniform when photon absorption occurs at different depth from surface, as shown in Fig C-12.

Below is a more detailed explanation.

With the same electric field, electrons and holes do have a different ionization probability α_e and α_h [14], the probability that an accelerated electron or hole creates another free electron-hole pairs via collision with bound electrons. At low electric field, the ionization ratio α_h/α_e is less than 0.1. Therefore, electrons are more likely to trigger an avalanche compared to holes. Fig C-12 shows the device breakdown probability distribution P_{BD} when the photons are absorbed in different positions of the depletion region, assuming a uniform electric field of 4×10^5 V/cm and based on the model in Ref [8] (relations derived by Oldham *et al.*). We report the equation here for the sake of convenience,

$$\begin{cases} \frac{dP_e}{dx} = (1 - P_e) \cdot \alpha_e \cdot (P_e + P_h - P_e P_h) \\ \frac{dP_h}{dx} = -(1 - P_h) \cdot \alpha_h \cdot (P_e + P_h - P_e P_h) \end{cases}$$

$$P_{BD} = P_e + P_h - P_e P_h$$

where P_e is a distribution of probability along x direction in the depletion region that an electron could trigger a sustainable avalanche, P_h is a distribution that a hole could trigger a sustainable avalanche and P_{BD} is the device breakdown probability distribution that an coming photon could trigger a sustainable avalanche.

In Fig C-12, P_{BD} is lower at the N+ and intrinsic layer interface because holes have a lower ionization probability. Photons that are absorbed in the top N+ layer will only have a small chance to trigger the SPAD, which means they will contribute less to the jitter distribution. Although there is absorption difference in the top N+ layer between the light trapping and control SPADs, their jitter distribution differences are not as large. However, if we reverse the doping polarity, the difference between the jitter distributions will be more obvious because almost all the minority carrier electrons absorbed in the top P+ layer then will contribute to SPAD jitter distribution.

Figure C-12 Simulated device breakdown probability when a photon is absorbed in the depletion region, which is denoted by the two vertical dashed lines. Shaded area correspond to region being nano-structured.

To make this point clear, we revise the manuscript as below:

Line 92: "However, the full-width tenth maximum (FWTM) and tail of the light-trapping SPAD are slightly better than that of control SPAD. This is because in light trapping SPADs, nano-structures are formed by etching silicon away, so fewer photons are absorbed in the top layer, as shown in the inset of Fig. 3(c)."

Line 96: "In our design, the top layer is heavily phosphorus doped to make holes as minority carriers. In Si, holes have a lower ionization probability compared to electrons. So chances for photo-generated carriers in the top layer to trigger an avalanche are smaller, and therefore they contribute less to the jitter distribution. The tail difference is expected to be larger if we reverse the doping polarity."

Comment 7:

"External Quantum Efficiency (EQE)" What exactly does this mean? Is this single-photon detection efficiency, or conversion of incident photons to electron-hole pairs?

Reply:

We thank the reviewer for this unclear point. External quantum efficiency (EQE) is the ratio of the photon-generated carriers collected by an SPAD to the number of external incident photons of a given wavelength. EQE is measured at 0 V in the manuscript. Photon detection efficiency (PDE) refers to the probability of triggering an SPAD given a single photon input, which is biased above the breakdown voltage. In the ideal scenario where exactly a single photon is injected, PDE is the multiplication of EQE and breakdown probability.

To make this point clear, we revise the manuscript as below:

Line 101: "External Quantum Efficiency (EQE) is used to evaluate absorption in an SPAD, which is defined as the ratio of photo-generated carriers collected by an SPAD to the number of external incident photons of a given wavelength. Measurements of EQE at 0V confirm our simulation results, as shown in Fig. 4(a)."

Comment 8:

“At wavelengths below 700 nm, the enhancement mainly comes from the anti-reflection effect of the nano-structure, with an additional contribution from nano-structure diffraction and effective collection of carriers.” I am not sure I understand why the carrier collection is more efficient when the nanostructure is present?

Reply:

We thank the reviewer for this comment. In Si, there will be more absorption close to surface when the wavelength is short. For example, the absorption length of 400 nm wavelength is only 100 nm. With nanostructures of inverse pyramids, photo-generated carriers near the surface diffuse less distance to the depletion region (300 nm on average), compared to control samples, where carriers have to diffuse 600 nm to reach the depletion. This helps enhance the collection efficiency because the diffusion distance is shorter.

In order to make it clear, we revise the manuscript as below:

Line 110: “For wavelengths whose absorption concentrates on the surface, the enhancement stems from on average less diffusion distance from the surface to the depletion region.”

Comment 9:

Figure 4 appears to show single-photon detection efficiency. These results have some potential from the point of view of improvement between devices, but I am very concerned that an SPDE of up to 25% is observed when a pulse containing 6.5 photons per pulse is incident. This surely will lead to highly distorted results as time-correlated single photon counting must operate only in the regime of low probability of detection, typically below 5% probability of a detection event when compared to the incident pulse rate. Otherwise, this will lead to highly unreliable SPDE and jitter results. (See “Pile-Up Effect” in W. Becker “Advanced Time-Correlated Single Photon Counting Techniques” Springer 2005). Essentially, the SPDE is the probability of a single-photon being measured, it is not the probability of a detection event if multiple photons are incident – that probability will always be higher than the SPDE. Generally, such an SPDE measurement will be valid if the probability of detection is very low, so a combination of high detection efficiency and

large number of photons per pulse cannot be valid conditions for an SPDE measurement.

Reply:

We thank the reviewer for this valuable comment. We agree that “time-correlated single photon counting must operate only in the regime of low probability of detection, typically below 5% probability of a detection event when compared to the incident pulse rate” is quite true in many applications, such as fluorescence analysis. But in characterizations of PDE and jitter performances of a SPAD in laboratory, the properties of input light are stable and can be characterized in detail, thus the multi-photon effect can be well corrected. We have done both numerical simulations and experiments to validate our multi-photon correction methods for PDE and jitter measurements, which can be found in Appendix I and Appendix II at the end of this letter. The two appendices show that high photon flux per pulse, or strictly, high trigger probability per pulse can yield the same PDE and jitter measurement results as low photon flux, when input light is well calibrated to take a multi-photon correction, or namely pile-up correction. Moreover, the appendices also show that the high DCR won't affect the PDE and jitter measurement results either. Our characterization method shows advantages to improved signal to noise ratio and reduced measurement time in the case of high DCR as in our scenario.

This comment also makes us double-check and improve the data processing programs in both PDE and jitter measurements. For PDE measurement, a bug is found and fixed, so that the PDE will not decrease at high excess voltage (the details could be found in Comment 5 to reviewer #2). With the corrected program and 20 μ m diameter devices, the saturated PDE of light trapping SPAD reaches 32%, compared to original 25% of 50 μ m device tested by old program. This result has been updated in Fig. 5 in revised manuscript (Fig. 4 in original) and the main text.

For jitter measurement, we also appreciate that you brought up the idea of pile-up effect, because our original measured jitter distribution did suffer from that and without any correction. To get correct results, we adopt the pile-up correction program as detailed in Appendix II, switch the laser wavelength to 940 nm and redo the measurements at 1 photon per pulse. At 940 nm wavelength, PDEs for both SPADs are lower (measured to be 15% for light-trapping SPADs and 5.5% for control SPADs). To demonstrate that pile-up correction method works, average 1, 6 and 20 photons per pulse were used to test the same light trapping SPAD. Fig C-8(a) (copied

below) shows the original measurement results, where pile-up effects are seen clearly, especially in the high photon flux scenarios. Then, by using a simple pile-up correction algorithm [11], the original jitter distribution is successfully restored, as shown in Fig C-8(b). The jitter distributions under 1, 6 and 20 photons per pulse overlap with each other quite well, which confirms that the pile-up correction works well under high photon flux. A FWHM of 25 ps, FWTM of 265 ps, and tail of 1380 ps can be read from the corrected jitter distributions. We have updated jitter distribution in Fig. 5(d) in the revised manuscript, to the pile-up corrected ones measured at 1 photon per pulse and at 940 nm wavelength.

We have also revised the manuscript accordingly. Explanations on jitter distribution measurements validity are located in the Appendix II: jitter distribution characterization.

Figure C-8 Pile-up effects in jitter distribution measurements and their corrections. **(a)** Measured jitter distributions from a light-trapping SPAD affected by pile-up. **(b)** Pile-up corrected jitter distributions.

Comment 10:

The DCR is 40MHz, which is exceptionally high for a silicon device, and will make measurement of SPDE very difficult. This is not mentioned. In fact, groups have shown even Ge-on-Si SPADs without guard rings with lower DCR than that described, (IEEE JOURNAL OF QUANTUM ELECTRONICS, VOL. 47, NO. 5, MAY 2011 and IEEE TRANSACTIONS ON ELECTRON DEVICES, VOL. 60, NO. 11, NOVEMBER 2013). Of

course, to achieve this, these groups cooled the detector and this allowed accurate SPDE measurements at a low photon number per pulse. These groups electrically gated the device during testing to reduce the overall measured count rate – was this done here? One highly plausible reason that the dark count rate of these all-Si devices are so high is that the structure is grown on SOI and not on an all-silicon substrate – an issue not discussed in this manuscript. Was an equivalent control sample grown on an all-silicon substrate measured?

Reply:

Thanks for pointing it out. The high DCR is mainly due to the absence of a guarding ring structure in standard SPAD, which shall be the next goal of our research. Here, in the revised version, we used gated quenching configuration with a gate width of ~ 10 ns and a repetition rate of 1 kHz. This helps suppress the overall measured dark count rate. We agree with you that the high DCR of light-trapping SPADs is partly due to growth on SOI substrates. DCR comparison between SPADs on a Si substrate and on an SOI substrate is demonstrated as in Fig C-13. SPADs on the SOI substrate have almost double the DCR than those on the Si substrate. There is a difference, yet they are within the same order of magnitude.

Figure C-13 Comparison of DCR between SPADs fabricated on an SOI substrate and a Si substrate.

Cooling measurements show that DCR decreases by 20% per 10°C, which suggests that there are tunneling breakdowns (e.g., trap-assisted tunneling breakdown) in the junction. We hypothesize that thin-junction SPADs of ~2 μm thickness often suffer from tunneling-related DCR due to narrower depletion widths. Similar phenomena can be found in Ref [5], [6].

The manuscript is revised as:

Line 136: "This (high DCR) may come from epitaxy on SOI substrates, lack of a guard ring structure, and tunneling in the thin-junction."

As mentioned in the previous comments, the validity of characterization under high DCR is demonstrated in Appendix I & II. Further, we re-characterized all the results with the 20 μm diameter SPADs. At the same excess voltage, this reduces DCR by ~5 times compared to 50 μm diameter SPADs.

Comment 11:

In conclusion this is an interesting manuscript, with evidence of efficiency enhancement by use of the nanostructure, and will merit interest from the single photon community if placed in a well-written manuscript. However, the paper is badly written, does not place the work in context, does not cite other work well and has poor description of device operation. The information provided along with the results of Figure 4 makes the measurement look incorrect. The conditions of high DCR and high photon number per pulse can make the efficiency result look higher than they actually are, and is a common mistake made by researchers new to the subject. I would suggest that the group cool the detectors to reduce the dark count rate and use a much lower number of photons per pulse. Doing this will give the community confidence that the results are correct. In addition, there are a number of other omissions in the manuscript (described above) that need attention. Overall, this paper is lacking adequate description of the measurement techniques and device physics. The Geiger mode results are, at best, questionable despite the possible enhancement seen in performance. I cannot recommend publication in its present form and suggest the authors revisit some of the Geiger-mode measurements, or direct a heavily edited manuscript in a different manner.

Reply:

Thanks for this insightful and valuable comment. Your consideration on our manuscript as “will merit interest from the single photon community if placed in a well-written manuscript” gives us great encouragement in both improving our work and revising the manuscript. We have spent more than three months to make a major revision to our manuscript. More references are cited and device physics is added according to your suggestions. What is more, as described previously, one bug in our data processing program is fixed and the multi-photon-correction methods for PDE and jitter distribution are further improved and double validated by both numerical simulations and experiments under high DCR and high photon flux conditions (See Appendix I and Appendix II for details). All single photon measurements are redone on a new batch of 20 μ m diameter devices and the results are processed by new data processing program and updated to the revised manuscript. We use 20 μ m diameter devices to reduce DCR instead of cooling, which is a very good suggestion but not available for us at this time and will be our future work. As we consider the measurement techniques as trivial and want to focus on the physics and result of light trapping, the measurement methods are only described in one subsection in the Method section. Instead we add two appendices at the end of this letter to validate measurement method instead of making the manuscript redundant.

We hope the new measurement results will convince you and the revised manuscript could really merit interest from the single photon community.

Once again, thank you very much for your comments and suggestions.

Appendix I: PDE measurement

Here we show the details of PDE measurements, and verify the validity of such PDE measurement method under high DCR and multiple photons per pulse by both numerical simulations and experiments.

PDE is characterized in a gated quenching configuration. The gate width is about 10 ns with a repetition rate of 1 kHz. An oscilloscope is used to monitor the SPAD's response in a 6.4 ns window. PDE is determined by the following equation:

$$1 - P_1 = (1 - P_0) * e^{-\mu \cdot PDE},$$

where P_1 denotes measured trigger probability per gate when laser is on, P_0 denotes measured trigger probability per gate while laser is off, and μ is the mean of photon number per laser pulse. The expression $\mu \cdot PDE$ could be intuitively understood as the mean number of photons detected by an SPAD, and $e^{-\mu \cdot PDE}$ is the Poisson probability that no photons are detected by the SPAD in this period. It is only when there are no photons in this pulse and no dark count in this period simultaneously, that we don't get SPADs triggered. Therefore, this equation takes into consideration of both multiple photons per pulse and high DCR scenarios.

We will then prove that this equation still holds when DCR is large. Suppose that in addition to current P_0 , DCR increases by k (e.g., increasing the ambient light). Then, in the new measurement with pile-up effect, we have the following:

$$P'_1 = P_1 + (1 - P_1)k$$

$$P'_0 = P_0 + (1 - P_0)k$$

$$e^{-\mu \cdot PDE'} = \frac{1 - P'_1}{1 - P'_0} = \frac{(1 - P_1)(1 - k)}{(1 - P_0)(1 - k)} = \frac{1 - P_1}{1 - P_0} = e^{-\mu \cdot PDE}$$

$$PDE' = PDE$$

which implies that increased DCR will not affect the PDE measurement results.

Figure A-1 Simulation of PDE versus average photon number per pulse and DCR. **(a)** A simulation that considers the pile-up effect for DCR and jitter distribution. Blue curve denotes the actual SPAD jitter distribution (ground truth). Red curve denotes distorted jitter distribution due to pile-up effect. Black curve denotes distorted DCR due to pile-up effect. Green curve denotes the difference between red curve (distorted jitter from pile-up effect) and black curve (distorted DCR from pile-up effect). **(b)** PDE characterized based on our method at different average photon number per pulse. **(c)** PDE characterized based on our method at different DCR.

Furthermore, we also draft a simulation code to simulate the pile-up influence on PDE characterization, given different photon numbers per pulse and different DCR. Fig A-1(a) shows an example from the simulation. The simulation settings are SPAD PDE of 40%, 1 photon per pulse on average and DCR of 51 MHz. The bin size is 5 ps, and the detection window is 6.4 ns. This scenario suffers from pile-up effect more seriously than our actual measurements. Then we applied the equation above by

using the data from the distorted DCR (black curve) and distorted jitter distribution (green curve) to characterize PDE. These two curves correspond to the data captured in the actual experiments. The results in Fig A-2(b) and (c) show that, the measured PDE remains the same at different photon numbers per pulse and at different DCRs. We have also estimated the measurement uncertainty of PDE in the aforementioned setting, which is $\pm 1.2\%$ out of 40% PDE.

Last but not least, we experimentally demonstrate the reliability of our PDE measurements under different scenarios. We choose a 20 μm diameter SPAD to further decrease DCR, compared to 50 μm diameter SPADs. In Fig. A-2(a), a light trapping SPAD with 20 MHz DCR is chosen. DCR is manually increased with introduction of the ambient light. Measured PDEs remain the same at increased DCR. In Fig A-2(b), we measure PDE at different photon numbers per pulse, namely 0.4, 0.6, 0.8, 3, and 6 photons per pulse. Measured PDEs also remain the same at increased photon flux. The two plots demonstrate that our method is reliable and valid in characterization of PDE under multiple photons per pulse and a high DCR.

Figure A-2 PDE measurements against different photon numbers per pulse and different DCRs. **(a)** PDEs of the light trapping SPAD (20 μm) measured at different DCRs. **(b)** PDEs of the light trapping SPAD (20 μm) measured at different photon numbers per pulse. Error bars denote measurement uncertainties in both figures.

In conclusion, by numerical simulations and experiments, our PDE measurement method can provide consistent results even when the average incident photon number reaches 6 per pulse, or when the DCR reaches 46 MHz.

Appendix II: jitter distribution measurement

Jitter distribution measurements need a large number of trigger events and is time consuming. To shorten the measurement time and get reliable results, we take the measurement at a large photon flux. But this will bring up the pile-up effect and correction is needed. Here we simulate and measure the influence of the pile-up effect, verify the pile-up correction algorithm, and apply the correction algorithm to the measurement results of SPADs under test.

The jitter distribution is obtained using a femtosecond laser of 940 nm wavelength and with a gated quenching configuration. Here, we simulate based on a jitter distribution of 25 ps FWHM, as shown by the blue curve in Fig A-1(a), which is close to the measured results. The simulation is based on a PDE of 40%. From Fig A-3(a), FWHM remains roughly the same (24-25 ps) even when the average photon number reaches 7 photons per pulse. In Fig A-3(b), FWHM remains the same at increasing DCR. This demonstrates that the FWHM measurement is reliable in our method, where we use 940 nm wavelength (PDE for light trapping SPADs is 15%) and 1 photon per pulse. Clearly, FWTM and FW(1%)M suffer from pile-up effect. A simple pile-up correction algorithm has been used [11] and the results are shown in Fig A-4.

Figure A-3 Simulation of jitter performance against different photon numbers per pulse and DCRs without pile-up correction. **(a)** Jitter distribution changes due to pile-up effect at increasing average photon number per pulse. **(b)** Jitter distribution changes due to pile-up effect at increasing DCR.

To test the pile-up effect and verify the pile-up correction algorithm, we measured the jitter distribution of a 20 μm diameter light trapping SPAD, whose PDE is

measured 15% at 940nm wavelength. Fig A-4(a) shows the measured results. Clearly, the pile-up effect is observed. However, after simple correction, the three curves match each other nicely, as shown in Fig A-4(b). Besides, the FWHM remains 25 ps even at 20 photons per pulse. This demonstrates that FWHM measurements are reliable even without pile-up correction.

Figure A-4 Measurements of jitter distribution against different photon numbers per pulse and pile-up correction results. **(a)** Measured jitter distributions at different average photon numbers per pulse. Pile-up effect is observed. **(b)** Jitter distribution after pile-up correction. They match each other well.

Based on the same principle, jitter distributions of both the light-trapping and control SPADs are characterized and corrected at 1 photon per pulse. Final results show that they are similar to each other in FWHM, but light trapping SPADs has a slightly smaller tail compared to control SPADs, which is predicted in the simulation. Typical figures of merit for light trapping SPADs are FWHM of 25 ps, FWTM of 265 ps and tail of 1380 ps, while control SPADs have FWHM of 25 ps, FWTM of 335 ps and tail of 1755 ps.

Figure A-5 Measured and corrected jitter distributions of the light-trapping and control SPADs.

Reference:

- [1] H. Savin *et al.*, "Black silicon solar cells with interdigitated back-contacts achieve 22.1% efficiency.," *Nat. Nanotechnol.*, vol. 10, no. May, pp. 1–6, 2015.
- [2] N. Ahmed *et al.*, "MOS Capacitor Deep Trench Isolation for CMOS image sensors," in *Technical Digest - International Electron Devices Meeting, IEDM*, 2015, vol. 2015–Febru, no. February, p. 4.1.1-4.1.4.
- [3] J. Ahn *et al.*, "7.1 A 1/4-inch 8Mpixel CMOS image sensor with 3D backside-illuminated 1.12 μ m pixel with front-side deep-trench isolation and vertical transfer gate," *Dig. Tech. Pap. - IEEE Int. Solid-State Circuits Conf.*, vol. 57, pp. 124–125, 2014.
- [4] A. S. Grove, O. Leistiko, and C. T. Sah, "Redistribution of acceptor and donor impurities during thermal oxidation of silicon," *J. Appl. Phys.*, vol. 35, no. 9, pp. 2695–2701, 1964.

- [5] P. Sun, R. Ishihara, and E. Charbon, "Flexible ultrathin-body single-photon avalanche diode sensors and CMOS integration," *Opt. Express*, vol. 24, no. 4, p. 3734, 2016.
- [6] M.-J. Lee, P. Sun, and E. Charbon, "A first single-photon avalanche diode fabricated in standard SOI CMOS technology with a full characterization of the device," *Opt. Express*, vol. 23, no. 10, p. 13200, 2015.
- [7] J. Ma *et al.*, "Simulation of a high-efficiency and low-jitter nanostructured silicon single-photon avalanche diode," *Optica*, vol. 2, no. 11, p. 974, 2015.
- [8] A. Gulinatti *et al.*, "Modeling photon detection efficiency and temporal response of single photon avalanche diodes," *Processing*, vol. 7355, pp. 1–17, 2009.
- [9] E. Charbon, "Single-photon imaging in complementary metal oxide semiconductor processes.," *Philos. Trans. A. Math. Phys. Eng. Sci.*, vol. 372, no. 2012, p. 20130100, 2014.
- [10] A. Lacaïta, S. Cova, M. Ghioni, and F. Zappa, "Single-Photon Avalanche Diode with Ultrafast Pulse Response Free from Slow Tails," *IEEE Electron Device Lett.*, vol. 14, no. 7, pp. 360–362, 1993.
- [11] P. B. Coates, "Pile-up corrections in the measurement of lifetimes," *J. Phys. E.*, vol. 5, no. 2, pp. 148–150, 1972.
- [12] A. Spinelli and A. L. Lacaïta, "Physics and numerical simulation of single photon avalanche diodes," *IEEE Trans. Electron Devices*, vol. 44, no. 11, pp. 1931–1943, 1997.
- [13] F. Panzeri, A. Gulinatti, I. Rech, M. Ghioni, and S. Cova, "Silicon SPAD with near-infrared enhanced spectral response," *SPIE Opt. + Optoelectron.*, vol. 8072, pp. 807206–807206–7, 2011.
- [14] T. Rang, "The impact-ionization coefficients of carriers and their temperature dependence in silicon," *Radioelectron. Commun. Syst.*, vol. 28, no. 5, p. 91, 1985.

Reviewers' Comments:

Reviewer #1:

Remarks to the Author:

The authors successfully addressed the points I raised; therefore I recommend this paper for the publication in the Nature Communications.

- The dark current with nanostructuring: They carried out extra experiment and clearly demonstrated the effect of nanostructuring on the PD dark current. And they also add in-depth discussion on this issue.
- The performance of a dielectric-metal-dielectric sandwich deep trench isolation (DTI) is calculated to represent its impact on the crosstalk among pixels.

Reviewer #2:

Remarks to the Author:

The authors properly replied to most of the reviewers' comments. As a result, the manuscript is much better than the first draft.

My main concern regards the impact of high photon fluxes on PDE and time jitter measurements. The straightforward method, followed by all the researchers working in the SPAD field, is to use the TCSPC (Time-Correlated Single-Photon Counting) technique and to limit the count rate in order to avoid distortions in the reconstructed waveforms. In detail, a rule of thumb is to limit the start-to-stop ratio in TCSPC measurements to 5% in order to have a distortion of less than 1%. This guarantees correct PDE and timing jitter estimations. Therefore, I think that the corrections described in appendix I and II of the rebuttal letter have to be avoided by acquiring the optical waveforms with low count rate. This is possible even when DCR is pretty high by working in gated mode in order to limit the impact of high DCR.

Additionally, I would suggest to include in the manuscripts more of the reply given to comments 2 and 3 of reviewer 2.

Reviewer #3:

Remarks to the Author:

The original manuscript contained some major flaws. I am satisfied that the authors have taken these numerous and important criticisms seriously and made a considerable effort to make the manuscript acceptable.

I am content to recommend publication in Nature Communications.

Reply to Reviewer #1

Comment 1:

The authors successfully addressed the points I raised; therefore I recommend this paper for the publication in the Nature Communications.

- The dark current with nanostructuring: They carried out extra experiment and clearly demonstrated the effect of nanostructuring on the PD dark current. And they also add in-depth discussion on this issue.
- The performance of a dielectric-metal-dielectric sandwich deep trench isolation (DTI) is calculated to represent its impact on the crosstalk among pixels.

Reply:

We are grateful to the reviewer for his/her recommendation on publication. Also, we would like to thank again the reviewer for his/her great effort in reviewing our manuscript and his/her insightful and valuable comments to improve the presentation of our work.

Reply to Reviewer #2

Comment 1:

The authors properly replied to most of the reviewers' comments. As a result, the manuscript is much better than the first draft.

My main concern regards the impact of high photon fluxes on PDE and time jitter measurements. The straightforward method, followed by all the researchers working in the SPAD field, is to use the TCSPC (Time-Correlated Single-Photon Counting) technique and to limit the count rate in order to avoid distortions in the reconstructed waveforms. In detail, a rule of thumb is to limit the start-to-stop ratio in TCSPC measurements to 5% in order to have a distortion of less than 1%. This guarantees correct PDE and timing jitter estimations. Therefore, I think that the corrections described in appendix I and II of the rebuttal letter have to be avoided by acquiring the optical waveforms with low count rate. This is possible even when DCR is pretty high by working in gated mode in order to limit the impact of high DCR.

Reply:

We would like to thank the reviewer for recognizing our revised manuscript as "much better than the first draft." We also appreciate the reviewer for informing us in details about the straightforward method, TCSPC technology, to measure PDE and timing jitter. We agree that limiting the start-to-stop ratio (trigger probability) to 5% or less can keep distortion less than 1%. This rule is very important in many TCSPC applications, such as fluorescence analysis. Undoubtedly, it makes sense to follow this rule in a SPAD characterization.

However, it is not practical to achieve this goal due to the current high DCR of 40MHz, even by working in gated mode as kindly suggested by the reviewer. Gated mode measurement with a 10ns gate width has already been performed in our previous tests. The minimum gate width is 2ns, due to the jitter distribution of the light trapping SPAD. At 2ns gate, we will still have a trigger probability of 8% to detect a dark count.

Therefore, in PDE and jitter distribution measurements, we have to use gating technology and the corrections described in appendix I and II in the previous response letter. The corrections are based on a basis that the properties of the input light in a SPAD characterization are stable and can be well known, thus the multi-photon effect and pile-up effect can be well corrected, which is different from many other TCSPC applications. Here, we have conducted some comparative experiments over several excess voltages to further prove the reliability of our characterization method. A thicker-junction SPAD mentioned in the previous response letter is chosen with a DCR of about 1MHz. It is measured under a gated width of 10ns (1% trigger probability for dark counts), first in low photon flux with total trigger probability following the 5% rule without any corrections, and then in high photon flux with corrections described in the appendix I and II in the previous response letter. The results are shown in **Figure 1**.

Figure 1 | Comparative measurements of PDE and timing jitter in low/high photon fluxes on a thicker-junction SPAD. In all figures, red curves stand for results in low photon flux which followed the 5% rule, without any corrections. Blue curves stand for results in high photon flux with corrections applied. **a**, Comparisons of total trigger probability per gate against different excess voltages in the two test conditions. **b**, Comparisons of PDE at 940nm wavelength against different

excess voltage in the two test conditions. **c**, Comparisons of FWHM timing jitter over different excess voltage in the two test conditions. **d**, Comparisons of timing jitter distribution at 2.8 V excess voltage in the two test conditions.

It should be pointed out that the structure of thicker-junction SPADs in the comparative tests are different from the light-trapping one. There is no strong light trapping so absorption is terrible at 940nm wavelength. Therefore both the PDE (due to absorption) and jitter performances (due to thicker junction of 3.5 μ m) are poor. However, this does not affect the comparisons of measurements in low or high photon flux. As shown in **Figure 1 a**, the trigger probability per gate in low photon flux are well below 5% over all tested excess voltages, thus the PDE or timing jitter estimated in low photon fluxes in **Figure 1 b, c, and d** have a distortion of less than 1% based on the rule of thumb. If we increase the photon flux so that the trigger probability per gate reaches 20% as shown in **Figure 1 a**, which is similar to the scenario when we characterize our light-trapping SPAD in the manuscript, the distortion will be quite large. However, as shown in **Figure 1 b, c, and d**, after applying the corrections described in the appendix I and II in the previous response letter, the estimated PDE and jitter distribution in high photon fluxes match very well with the results from traditional low photon fluxes.

We admit that it is a compromise to use large trigger probability and corrections to characterize our SPADs. However, this method has been proven by the theoretical derivation, numerical simulations and comparative experiments. The experimental results are repeatable and reliable. We really appreciate the reviewer's suggestion and hope that this further experiment will bring up confidence in our characterization results.

Comment 2:

Additionally, I would suggest to include in the manuscripts more of the reply given to comments 2 and 3 of reviewer 2.

Reply:

We thank the reviewer for pointing this out. It definitely helps clarify confusion between simulation and experiment. Therefore we revised the manuscript accordingly as below.

Line 73: The breakdown voltage is lower than the design for two reasons. One is dopant diffusion during thermal oxidation at 1000 °C, which reduces the depletion region thickness (see Supplementary Fig. 3); the other is due to edge effect and lack of a guard ring, where the breakdown probability is measured to be 20% higher on the edge.

Line 92: A simulation of jitter distribution has been performed to make a qualitative comparison, assuming uniform electric field in the 1.2 μm thick depletion region. The electric field is chosen to be 4.5×10^5 V/cm, which corresponds to a FWHM jitter of 25 ps and an excess voltage that is 35% of the breakdown voltage. The simulation conditions are set to match our experiment result (25ps, 30% excess-breakdown voltage ratio).

Supplementary Figure 3 is added for better understanding.

Again, we would like to thank the reviewer for his/her great effort in reviewing our manuscript and his/her insightful and valuable comments which are certainly helpful to improve the presentation of our work.

Reply to Reviewer #3

Comment 1:

The original manuscript contained some major flaws. I am satisfied that the authors have taken these numerous and important criticisms seriously and made a considerable effort to make the manuscript acceptable.

I am content to recommend publication in Nature Communications.

Reply:

We are grateful to the reviewer for his/her recommendation on publication. Also, we would like to thank again the reviewer for his/her great effort in reviewing our manuscript and his/her insightful and valuable comments to improve the presentation of our work.

Reviewers' Comments:

Reviewer #2:

Remarks to the Author:

With the last additional information, the authors proved with comparative experiments that their device characterization and data processing is reliable.

Therefore, I now recommend the publication of this paper.

Reply to Reviewer #2

Comment 1:

With the last additional information, the authors proved with comparative experiments that their device characterization and data processing is reliable.

Therefore, I now recommend the publication of this paper.

Reply:

We thank the reviewer for his/her recommendation on publication. Also, we are grateful to the reviewer for his/her great effort in reviewing our manuscript and his/her insightful comments to improve the presentation and reliability of our work.